# Size-resolved process understanding of stratospheric sulfate aerosol following the Pinatubo eruption

Allen Hu[1,*], Ziming Ke[2,*], Xiaohong Liu[1], Benjamin Wagman[3], Hunter Brown[3], Zheng Lu[1], Mingxuan Wu[4], Hailong Wang[4], Qi Tang[2], Diana Bull[3], and Kara Peterson[3], and Shaocheng Xie[2]

[1]Department of Atmospheric Sciences, Texas A&M, College Station TX, 77843, United States
[2]Lawrence Livermore National Laboratory, Livermore CA, 94550, United States
[3]Sandia National Laboratories, Albuquerque NM, 87185, United States
[4]Pacific Northwest National Laboratory, Richland WA, 99352, United States

* These authors contributed equally to this work.

*Correspondence to*: Xiaohong Liu (xiaohong.liu@tamu.edu)

**Abstract.** Stratospheric sulfate aerosol produced by volcanic eruptions plays important roles in atmospheric chemistry and the global radiative balance of the atmosphere. The simulation of stratospheric sulfate concentrations and optical properties is highly dependent on the chemistry scheme and microphysical treatment. In this work, we implemented a sophisticated gas-phase chemistry scheme (full chemistry, FC) and a 5-mode version of the Modal Aerosol Module with Prognostic Stratospheric Aerosol (MAM5-PSA) for the interactive treatment of stratospheric sulfate aerosol in the Department of Energy's Energy Exascale Earth System Model version 2 (E3SMv2) model to better simulate the chemistry-aerosol feedback following the Pinatubo eruption, and to compare it against a simulation using simplified chemistry (SC) and the default 4-mode version of the Modal Aerosol Module (MAM4). MAM5-PSA experiments were found to better capture the stratospheric sulfate burden from the eruption of the volcano to the end of 1992 as compared to the High-resolution Infrared Sounder (HIRS) observations, and the formation of sulfate in MAM5-PSA with FC (with an additional OH replenishment reaction) was significantly faster than in MAM4 with FC . Analyses of microphysical processes indicate that more sulfate aerosol mass was generated in total in FC experiments than in SC experiments. MAM5-PSA performs better than MAM4 in simulation of aerosol optical depth (AOD); AOD anomalies from the MAM5-PSA experiment have better agreement with observations. The simulated largest changes in global mean net radiative flux at the top of the atmosphere following the eruption were about -3 W/m$^2$ in MAM5-PSA experiments and roughly -1.5 W/m$^2$ in MAM4 experiments.

## 1 Introduction

Explosive volcanic eruptions can inject material and gas-phase aerosol precursors into the stratosphere, affecting atmospheric chemistry and radiation and often leading to significant global-scale cooling at the surface (Yang and Schlesinger, 2002; Parker et al., 1996). Many major volcanic eruptions have been associated with dramatic drops in temperature, famine and even the absence of a summer season (Robock, 2000), though Mt. Pinatubo was the first major eruption to be observed with modern instruments. Background stratospheric aerosol during volcanically quiescent periods have optical depths of less than 0.01 and a total stratospheric burden of roughly 0.1 Tg (Sheng et al., 2015; Deshler, 2008), and mainly consist of sulfate formed through the oxidation of sulfur dioxide ($SO_2$) and carbonyl sulfide (OCS) emitted at the Earth's surface (Sheng et al., 2015). These background aerosols are often overshadowed by sulfate aerosols produced by volcanic eruptions, which are formed from the oxidation of $SO_2$ into sulfuric acid and subsequent nucleation and condensation to form sulfate aerosol in the stratosphere (Mills et al., 2016). These particles have a lifespan on the order of years, so it is rare for the stratosphere to be unperturbed by volcanic emissions (Seinfeld and Pandis, 2016). Eventually, these stratospheric aerosols are removed by entering the troposphere through sedimentation processes and cross tropopause air mixing (Kremser et al., 2016).

The eruption of interest in this study is the Mt. Pinatubo eruption on June 15, 1991. It was one of the most powerful volcanic eruptions in recent decades and occurred after the onset of various modern observation techniques. The amount of $SO_2$ injected into the atmosphere was likely in the range of 15 to 18 Tg (Carn, 2022; Neely Iii, 2016; Aubry et al., 2021). A significant cooling of approximately 0.5 degree at the Earth's surface was observed afterwards due to

the strong scattering of solar shortwave radiation by sulfate aerosol (Parker et al., 1996). In recent years interest in geoengineering as a countermeasure against the effects of climate change has been growing (Crutzen, 2006), with a concept that is analogous to that of a volcanic eruption; sulfate aerosol, formed from $SO_2$ that is deliberately injected into the stratosphere, could lead to significant temperature decreases at the Earth's surface. It is therefore essential to examine volcanic eruptions for better modeling of future eruptions and better understanding of the effects of geoengineering.

Numerous previous studies have examined volcanic effects on climate and possible influencing factors such as the location and timing of the eruption and total emissions. Using statistical emulation of output of the UK Met Office Unified Model coupled with the UK Chemistry and aerosol scheme (UK-UMCA), Marshall et al. (2019) found that for large $SO_2$ emissions of 10 to 100 Tg, the e-folding decay time for the global sulfate aerosol burden is most dependent on the latitude of the eruption, while the integrated global mean stratospheric AOD and net radiative forcing is most dependent on the total mass of the $SO_2$ emissions. Zhuo et al. (2024) used the Whole Atmosphere Community Climate Model Version 6 (WACCM6) to simulate volcanic eruptions on a similar scale to Pinatubo from Central America and Iceland under different ENSO initial states, finding that initial atmospheric conditions control the meridional transport of sulfur and halogens in the first month after the eruptions as well as further modulating the latitudinal distribution of sulfate aerosols, halogens, volcanic forcing and impacts. Toohey et al. (2019) analyzed ice-core-derived volcanic stratospheric sulfur injections and Northern Hemisphere summer temperature reconstructions from tree rings, finding that in proportion to their estimated stratospheric sulfur injection, extratropical explosive eruptions since 750 CE have produced stronger hemispheric cooling than tropical eruptions; stratospheric aerosol simulations with the MAECHAM5-HAM model suggested that this was due to the enhanced radiative impact associated with the relative confinement of aerosol to a single hemisphere.

Previous studies regarding stratospheric aerosols utilize different methods to simulate aerosol properties and microphysics. The simplest and most direct approach is to prescribe stratospheric aerosol properties or burdens using climatology derived from observations. For example, Zhuo et al. (2021) produced volcanic forcing using the Easy Volcanic Aerosol (EVA) module, which directly generated stratospheric aerosol optical properties for the given volcanic emissions. More sophisticated methods primarily include bulk schemes, modal schemes, and sectional schemes (Liu, 2023). Bulk schemes are the simplest, where aerosols species are not divided into bins or modes, and properties such as size distribution are prescribed. CNRM-ESM2-1 (an Earth system model developed by the Centre National de Recherches Météorologiques) – one of the models used in Tilmes et al. (2022) for a model intercomparison project for geoengineering – prescribed stratospheric aerosol size distributions with no interactive aerosol microphysics. Gao et al. (2023) used the GFDL Earth System Model version 4.1 (GFDL ESM4.1) to prognostically simulate stratospheric sulfate aerosol concentrations for volcanic eruptions but aerosol size was prescribed, with a sulfate dry effective radius of 0.166 μm, 0.25 μm, 0.4 μm or 1 μm in different sensitivity experiments. Modal schemes typically divide the aerosol population according to the modes conventionally used to describe aerosol size distributions (i.e., Aitken mode, accumulation mode, coarse mode, and nucleation mode in some schemes), with each

mode having its own prescribed standard deviations of lognormal distributions while mass and number concentrations of aerosols in each mode are predicted (Mills et al., 2016; Visioni et al., 2023). Mills et al. (2016) used WACCM's three mode version of the Modal Aerosol Module (MAM3) to simulate the Pinatubo eruption by altering the parameters (e.g., standard deviation) of the coarse mode, but in doing so also influenced the simulation of unrelated coarse mode aerosols such as sea salt and dust, as the aerosols in each mode are treated as internal mixtures rather than being separated by species. Brown et al. (2024) simulated the Pinatubo eruption in E3SMv2 using a four mode version MAM (MAM4), with modifications to the default MAM4 used in E3SMv2. Bin schemes (or sectional schemes) divide aerosols into more categories than modal schemes, providing greater resolution with the drawback of additional computational cost. Vattioni et al. (2019) used a sectional aerosol module which was capable of handling aerosols in 40 different size bins to better represent accumulation-mode $H_2SO_4$ of stratospheric aerosol and their direct injection into the atmosphere as opposed to injections of $SO_2$. Laakso et al. (2022) simulated sulfur injection in the stratosphere using the Sectional Aerosol module for Large Scale Applications (SALSA) which utilized 10 size bins.

Recent studies of volcanic eruptions or stratospheric sulfate related to geoengineering have utilized different approaches with regards to stratospheric chemistry. Zhuo et al. (2021) omitted chemistry of sulfate formation entirely, with $SO_2$ emissions directly being converted into aerosol optical properties through empirical calculations. Some studies used a simplified chemistry scheme. In the SALSA1 scheme introduced in Kokkola et al. (2008) and implemented in Bergman et al. (2012), prescribed hydroxyl radical (OH) concentrations were used for stratospheric chemistry. Kleinschmitt et al. (2018) used a prescribed $SO_2$ to $H_2SO_4$ conversion rate in the Sectional Stratospheric Sulfate Aerosol (S3A) module to simulate $SO_2$ injections. Kleinschmitt et al. (2017) reported a doubling of $SO_2$ lifetime in the stratosphere when OH is prognostically calculated for major injections of $SO_2$, such as the Pinatubo eruption, but did not quantitatively compare scenarios with OH prescribed or not prescribed. Studies that used WACCM, such as Mills et al. (2016) and Visioni et al. (2019), considered the key chemical reactions within the stratosphere with a prognostic treatment of OH for the oxidation of $SO_2$ to form sulfate aerosol.

This paper builds upon the work of Mills et al. (2016) and Brown et al. (2024) by adding a unique stratospheric sulfate coarse mode and by considering the full chemistry in E3SM, to simulate the burden, AOD, and radiative effect of volcanic aerosols. Brown et al. (2024) also simulated the Pinatubo eruption using E3SMv2, the same version of the earth system model as this study. Their prognostic aerosol simulations use the model's default "simple" chemistry, the same as our SC experiments. One of their experiments, E3SMv2-PA, used the default MAM4 (i.e., nearly identical to our MAM4SC experiment). Their other prognostic aerosol experiment, E3SMv2-SPA, used a revised MAM4, which treated dust, sea salt and stratospheric sulfate using the same coarse mode parameters, such as size range and geometric standard deviation, specifically chosen to best represent stratospheric sulfate properties, resulting in erroneous coarse mode dust and sea salt aerosol concentrations. To avoid these problems, this study establishes a fifth aerosol mode (MAM5-PSA), the stratospheric coarse mode, to specifically handle stratospheric sulfate. The coarse mode remains the same as the original MAM4 to handle dust and sea salt. Our aim is to (1) examine the differences in simulated sulfate aerosol burden after the Pinatubo eruption with and without the addition of the stratospheric coarse

mode to MAM; (2) examine the differences in sulfate between using E3SM's default "simple" chemistry and a sophisticated "full" chemistry treatment; and (3) quantify the microphysical processes involved in sulfate chemistry and modal aerosol. It is worth noting that changes in stratospheric ozone following volcanic eruptions play a significant role in both stratospheric chemistry and the impacts on temperature, but it is not the focus for this paper.

The model set-up, including aerosol module, chemistry settings, emissions, nudging, and observational data are described in Section 2. Section 3.1 presents the simulated spatial distribution and total burden of stratospheric sulfate aerosols following the Pinatubo eruption, and Section 3.2 engages in process analyses of the sulfate aerosols and discusses their growth processes in the different experiments. Section 4 presents the conclusion of our research.

## 2 Methodology

### 2.1 Model overview

The model utilized in this study is E3SMv2, which runs roughly twice as fast relative to E3SMv1. The model contains further changes to the dynamical core, the dynamical grid and parameterization column grid, and atmospherics physics and chemistry (Golaz et al., 2022). By default, E3SMv2 treats aerosols through an enhanced version of MAM4 with all aerosol types appearing in the coarse mode (Liu et al., 2016), in which the four lognormal size modes represent the Aitken mode, accumulation mode, coarse mode and primary carbon mode respectively (the primary carbon mode is not directly relevant for our study of stratospheric sulfate aerosol). The model also simulates aerosol microphysical processes relevant to stratospheric sulfate, including condensation, nucleation, coagulation, and water uptake. The experiments are run under the "ne30pg2" resolution, with a grid spacing of approximately 110 km for dynamics and about 165 km for physics. Vertically, it has 72 layers of varying thicknesses, with the top at roughly 60 km altitude (Golaz et al., 2022). The emissions and wet and dry removal of aerosols were described in Wang et al. (2020).

In E3SMv2 MAM4, the coarse mode was primarily intended for coarse sea salt and dust, as well as other aerosol species resuspended from raindrop evaporation, with a geometric standard deviation of 1.8, and stratospheric sulfate aerosol cannot enter the coarse mode through renaming. Renaming is a process in which aerosol particles that grow larger than a given threshold via condensation and coagulation are transferred from one mode to another. To avoid these problems, this study establishes a fifth aerosol mode (MAM5-PSA), the stratospheric coarse mode, to specifically handle stratospheric sulfate. The coarse mode remains the same as the original MAM4 to handle dust and sea salt. The newly added stratospheric coarse mode follows the same size parameters as the revised coarse mode in Mills et al. (2016) and Brown et al. (2024) to better reflect stratospheric sulfate lifespans, but handles stratospheric sulfate separately from coarse mode sea salt and dust. A portion of sufficiently large accumulation mode sulfate aerosols are permitted to transfer to the stratospheric coarse mode through renaming. It should be noted that the upper limit for the accumulation mode size range is larger than the lower limit for the stratospheric coarse mode size range, leading to potential size overlaps between the two modes' aerosol populations.

To understand the mechanisms of simulated sulfate aerosol in MAM5-PSA, we carry out process analyses for (a) NUCL (nucleation), (b) COAG (coagulation) and (c) RNMaa (renaming from Aitken mode to accumulation mode), and (d) RNMasc (renaming from accumulation mode to stratospheric coarse mode) (Fig. 1(a)). Condensation processes for each of the modes have also been included, with the positive values representing the transition of mass from the gaseous phase to the aerosol. Each of these processes have their physical rules within MAM, and represent an actual physical process with the exception of renaming, which represents a computational step:

- NUCL, or nucleation, is responsible for the formation of aerosols starting from the Aitken mode (nucleation mode aerosols are not represented in MAM).

- COAG, or coagulation, represents two smaller particles colliding to form a larger particle. MAM considers the intramodal and intermodal coagulation of the Aitken, accumulation and primary carbon modes. Intramodal coagulation reduces the number of the mode but leaves the mass unchanged. Intermodal coagulation between these modes and the coarse mode is not considered in MAM.

- AitkenCond, or condensation in the Aitken mode, leading to aerosol mass increase in the Aitken mode population. No matter how much mass is gained from condensation, the aerosols cannot grow into the accumulation mode directly, it MUST undergo renaming.

- AccumCond and StratoCoarseCond, condensation for the accumulation mode and stratospheric coarse mode, work similarly.

- Periodically, the size of the Aitken mode aerosol population is checked. If a specified threshold is reached (due to mass increase from the condensation and coagulation processes), then aerosol number and mass are transferred from the Aitken mode to the accumulation mode (so called renaming, RNMaa). A similar renaming process transfers aerosol number and mass from the accumulation mode to stratospheric coarse mode (RNMasc). No other microphysical process is allowed to produce stratospheric coarse mode sulfates, so this is their only source. We note that renaming is a computational step in MAM, and does not correspond to a physical process.

We also improve the E3SMv2 to output these tendencies in 2D after the 3D tendencies are vertically integrated.

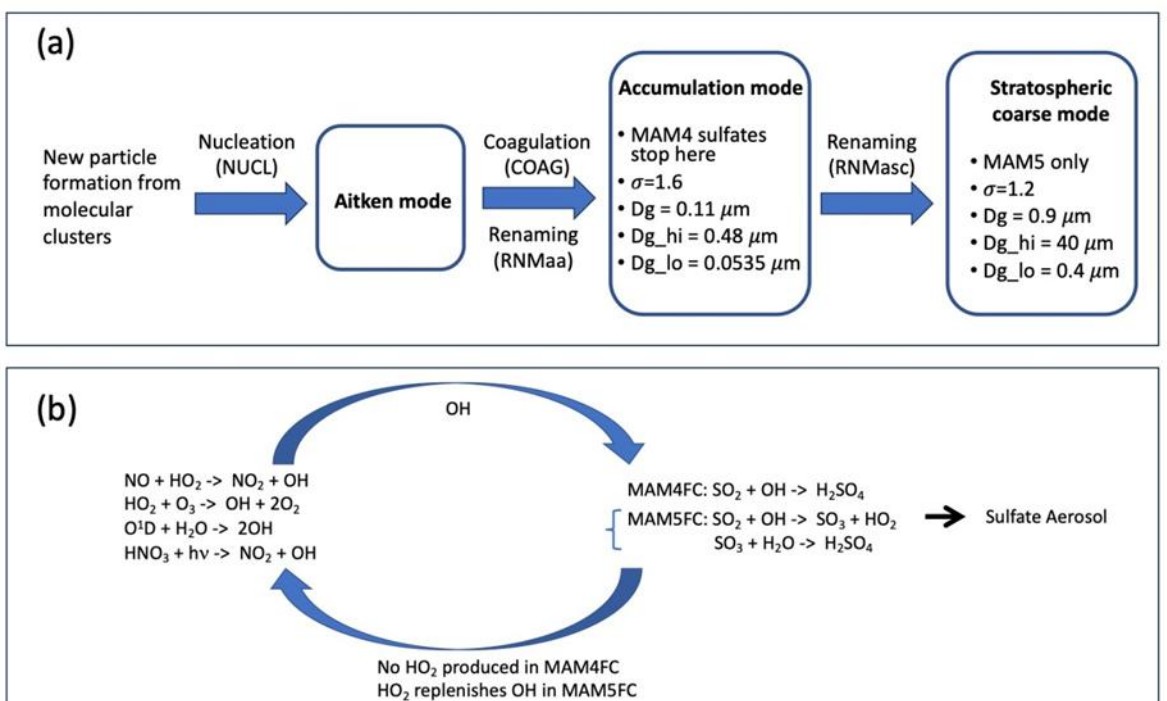

**Figure 1: (a) Flowchart depicting the aerosol modes and analyzed microphysical processes for the formation of stratospheric sulfate in our experiments. The parameters for the lognormal size distribution of the accumulation mode and stratospheric coarse mode are shown. σ represents the geometric standard deviation of aerosol number concentration (a smaller standard deviation corresponds to a longer atmospheric lifespan). Renaming is a technical step within E3SM where aerosols that grow sufficiently large can transfer from one mode to another. Dg, Dg_hi, and Dg_lo represent the initial values of the geometric dry mean diameter and its upper and lower limits, respectively. (b) Chart that depicts the differences in the full chemistry (FC) between MAM4 and MAM5-PSA. In MAM4FC, the oxidation of $SO_2$ by OH radicals directly produces $H_2SO_4$ with no byproducts. In MAM5FC, this reaction instead produces $HO_2$ and $SO_3$. $SO_3$ eventually converts into sulfuric acid, while $HO_2$ is crucial in the replenishment of OH concentrations.**

Note that the geometric dry mean diameter (Dg) values are prognostically calculated and can become larger or smaller than the initial prescribed Dg value given in Fig. 1(a) depending on microphysical processes. Dg_hi and Dg_lo are the upper and lower limits for the geometric dry mean diameter. These values are comparable to those used in Mills et al. (2016) so as to maintain parity. The reduction in the geometric standard deviation (from 1.6 in the accumulation mode to 1.2 in the stratospheric coarse mode) is a method previously used in Wu et al. (2020) to lengthen the lifespan of aerosols.

## 2.2 Chemical mechanisms

By default, E3SMv2 uses a simplified set of chemical species and reactions (hereafter referred to as "simple chemistry" or "SC") that omits many less important species and reactions to save computational costs. Tracer species in the model have prognostically evolving concentrations and include $H_2O_2$, dimethyl sulfide (DMS), sulfuric acid ($H_2SO_4$), $SO_2$, secondary organic aerosol precursor gases, and aerosols. Prescribed species such as oxygen, ozone, OH, and $HO_2$ have fixed, predetermined concentrations. Photolysis is restricted to $H_2O_2$ (and is simplified to produce

no product); beyond that, there are only six other reactions (note: R1 and R2 have different reaction rates, with the latter requiring an additional chaperone molecule (Emmons et al., 2010)):

$$DMS + OH \ \rightarrow SO_2 \tag{R1}$$

$$DMS + OH \ \rightarrow \ 0.5SO_2 + 0.5HO_2 \tag{R2}$$

$$DMS + NO_3 \rightarrow SO_2 + HNO_3 \tag{R3}$$

$$HO_2 + HO_2 \ \rightarrow \ H_2O_2 \tag{R4}$$

$$H_2O_2 + OH \ \rightarrow \ H_2O + HO_2 \tag{R5}$$

$$SO_2 + OH \ \rightarrow H_2SO_4 \tag{R6}$$

To better simulate the stratospheric chemistry following a volcanic eruption – albeit at greater computational cost (roughly twice that of SC) – a "full chemistry" (FC) setup is added through replacing the chemistry preprocessor. It is similar to the Model of Ozone and Related chemical Tracers (MOZART) chemistry scheme (Emmons et al., 2010) used in WACCM (Mills et al. (2016); the original work for E3SM is based on Wu et al. (2022), with some relevant stratospheric gas phase reactions added in. This scheme can handle many organic compounds that are ignored in the simple chemistry, containing 16 photolysis reactions and 41 stratospheric reactions that influence the concentrations of many radical species in the stratosphere. Most importantly, OH radicals (as well as other oxidants) are prognostically calculated rather than using prescribed values. This allows the model to represent the localized depletion of these species due to the injection of large amounts of $SO_2$, bottlenecking aerosol formation rates.

With the definitions of SC and FC described above, we consider combinations of MAM4SC, MAM4FC, MAM5SC, and MAM5FC in this study (Table 1). One further change is made for MAM5FC: for the oxidation of $SO_2$ to form sulfuric acid, the reaction in MAM4FC is $SO_2 + OH \ \rightarrow H_2SO_4$, whereas the reactions in MAM5FC are

$$SO_2 + OH \ \rightarrow SO_3 + HO_2 \tag{R7}$$

$$SO_3 + H_2O \rightarrow H_2SO_4 \tag{R8}$$

$HO_2$ participates in the replenishment of OH radicals in the stratosphere in MAM5FC, a process that is significantly slower in MAM4FC due to the above reaction not producing $HO_2$ (Fig. 1(b)).

### 2.3 Experimental set-up

To examine the differences between MAM4/MAM5-PSA and simple/full chemistry in E3SMv2's simulation of Pinatubo, one experiment is performed for each permutation. Coupled Model Intercomparison Project Phase 6 (CMIP6) Diagnosis, Evaluation and Characterization of Klima (DECK) emissions files are used with the exception of volcanic emissions, which consists of VolcanEESM emissions regridded and merged into CMIP6 $SO_2$ emissions (Brown et al., 2024). Pinatubo eruptions are assumed to have occurred on June 15 1991, with 10 Tg of $SO_2$ evenly emitted between 18 and 20 km altitude (Mills et al., 2016). This corresponds to the mass detected in the stratosphere by the TIROS Operational Vertical Sounder and Total Ozone Mapping Spectrometer 7-9 days after the beginning of the eruption (Guo et al., 2004). This is smaller than the observed $SO_2$ emissions because E3SM does not consider volcanic ash or water vapor emissions nor consequent ice sequestration of $SO_2$ and fallout of ash and ice particles. Sea

ice and sea surface temperatures are prescribed according to CMIP6 DECK datasets. Each simulation starts from January 1, 1991, with more than six months as spin up time. Only the meridional and zonal winds were nudged to
260 MERRA-2 reanalysis leaving the temperature and humidity to dynamically respond.

**Table 1: List of numerical simulations.**

| Experiment name | Aerosol module | Chemistry |
|---|---|---|
| MAM4SC | MAM4 | Simple chemistry |
| MAM4FC | MAM4 | Full chemistry |
| MAM5SC | MAM5-PSA | Simple chemistry |
| MAM5FC | MAM5-PSA | Full chemistry (with alterations) |

**2.4 Observational data**

AVHRR monthly observations of AOD for the years of 1989-1993 were used for model evaluation. Observed AOD values at 600 nm were downloaded from https://www.ncei.noaa.gov/data/avhrr-aerosol-optical-thickness/access (last access: June 5 2024) (Zhao, 2022). The wavelength of observation is different from E3SMv2 which outputs AOD at 550 nm wavelength, but the discrepancy caused by the difference in wavelength can be considered negligible. The website provides retrieved monthly AOD with a spatial resolution of 0.1°×0.1°, calculated using AVHRR's daily
orbital observation of top-of-atmosphere reflectance over the oceans. E3SM output AOD that is used to compare with AVHRR results is masked according to the pixels that were successfully observed by AVHRR and regridded to be the same resolution. The Global Space-based Stratospheric Aerosol Climatology (GloSSAC) is a global and gap-free data set of zonally averaged optical properties of stratospheric aerosols (focused on aerosol extinction coefficient at 525 and 1020 nm) from 1976–2018 (Thomason et al., 2018). Here, we use the updated version 2 from Kovilakam et
al. (2020). Sulfate burdens derived from HIRS observations according to Baran and Foot (1994) were used for model comparison. Data from the balloon-borne University of Wyoming optical particle counters (WOPC) (Deshler et al., 1993; Deshler et al., 2003) is converted from the original particle number size distributions to particle volume size distributions for comparisons with model outputs.

**3 Results**

**3.1 Simulated concentrations of stratospheric sulfate**

Figure 2 presents the simulated horizontal distributions of sulfate aerosol concentrations at 53 hPa at 3, 9 and 15 days after the eruption of Pinatubo respectively for the four experiments. The overall distribution pattern is similar between all four experiments, with sulfate aerosol spreading westwards from Mt. Pinatubo in the Philippines following the eruption because of the Quasi-Biennial Oscillation. After 3 days, the sulfate reaches India; after 9 days, it passes over
285 Africa and reaches the eastern coast of the Atlantic Ocean; after 15 days, it reaches Central America. Within this time

period, northward and southward transport is both relatively weak, and the sulfate mostly remains between 30 °S and 30 °N (excluding background sulfates present prior to the eruption).

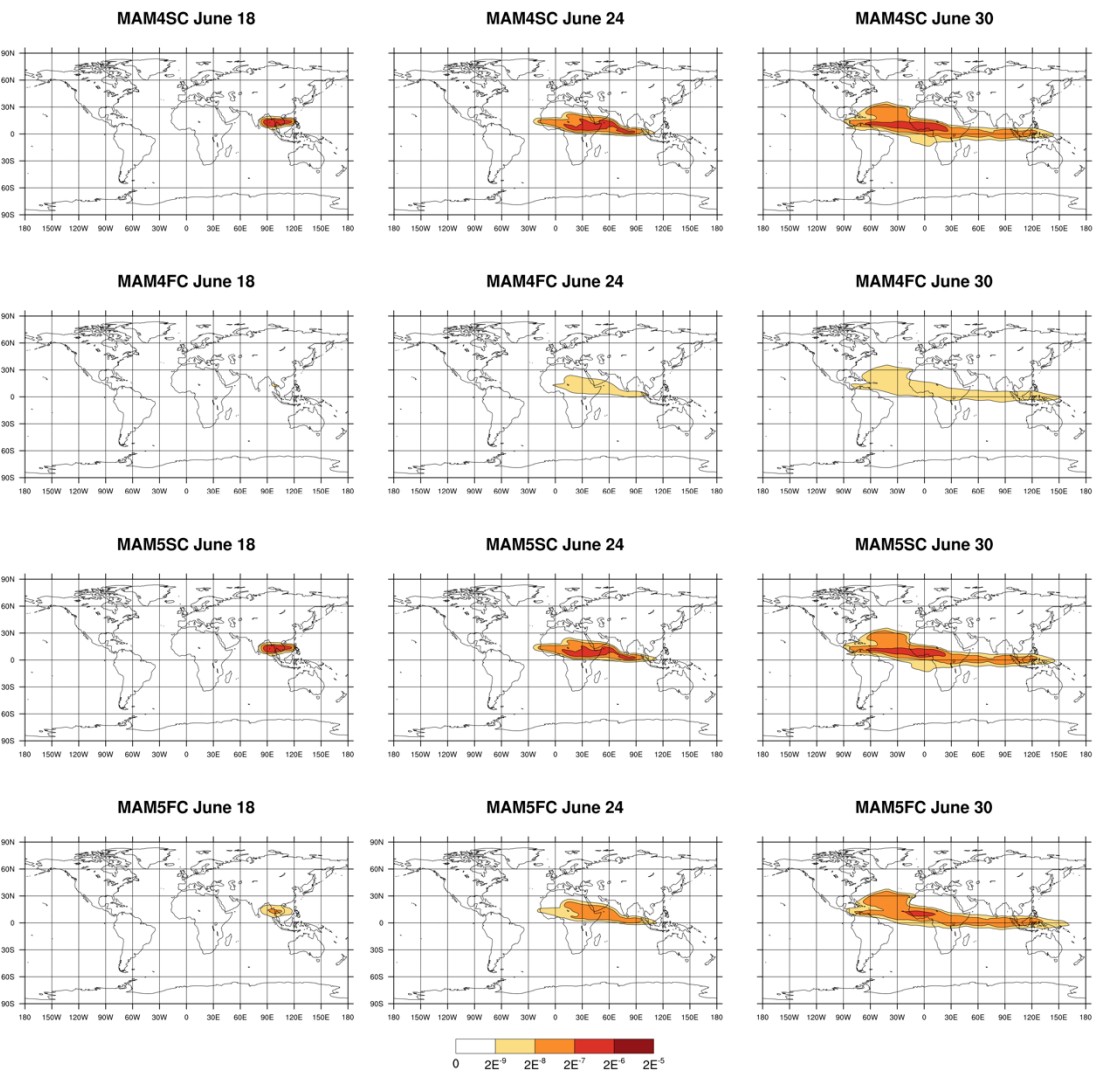

**Figure 2: Simulated distributions of sulfate aerosol concentrations (kg/kg) at 53 hPa for days 3, 9 and 15 after the eruption of Pinatubo, respectively, for experiments MAM4SC (first row), MAM4FC (second row), MAM5SC (third row), and MAM5FC (bottom row).**

For both MAM4 and MAM5-PSA simulations, sulfate concentrations are generally higher in SC than in FC. In these experiments, sulfate formation is not hindered by a lack of OH because OH concentration is prescribed and not depleted. On June 18, 3 days after the eruption, there is a substantial difference in sulfate concentrations near the origin of the plume. In MAM4SC and MAM5SC, the concentrations reach above $10^{-6}$ kg/kg. There is no replenishment mechanism for OH radical in MAM4FC, hindering sulfate formation, so the peak value in MAM4FC is below $10^{-8}$ kg/kg. In MAM5FC, OH can be depleted but a replenishment mechanism is also present, and the highest concentration is between $10^{-8}$ kg/kg and $10^{-7}$ kg/kg. These differences between SC and FC weaken over time as the plume spreads

and encounters more OH to participate in the oxidation of $SO_2$ in FC. On June 30, the peak concentration in MAM4FC is between $10^{-8}$ and $10^{-7}$ kg/kg, while in the other three experiments it is between $10^{-7}$ and $10^{-6}$ kg/kg.

Figure 3 depicts the vertical profiles of tropical sulfate concentrations for the Aitken and accumulation modes (or the sum of accumulation and stratospheric coarse modes for MAM5-PSA, as MAM4 cannot rename accumulation mode into coarse mode) in each of the four experiments. In all four experiments, the initial sulfate burden before July consists of mostly Aitken mode aerosols with peak concentrations on the order of $10^{-9}$ kg/kg. Sulfate aerosol either leaves the Aitken mode or is removed from the stratosphere within the span of several weeks in SC experiments, while sulfate persists past October in FC runs. Soon after July 1991, accumulation mode sulfate in MAM4 and accumulation mode plus stratospheric coarse mode sulfate in MAM5-PSA become dominant until they diminish back to background levels, with peak concentrations on the scale of $10^{-7}$ kg/kg.

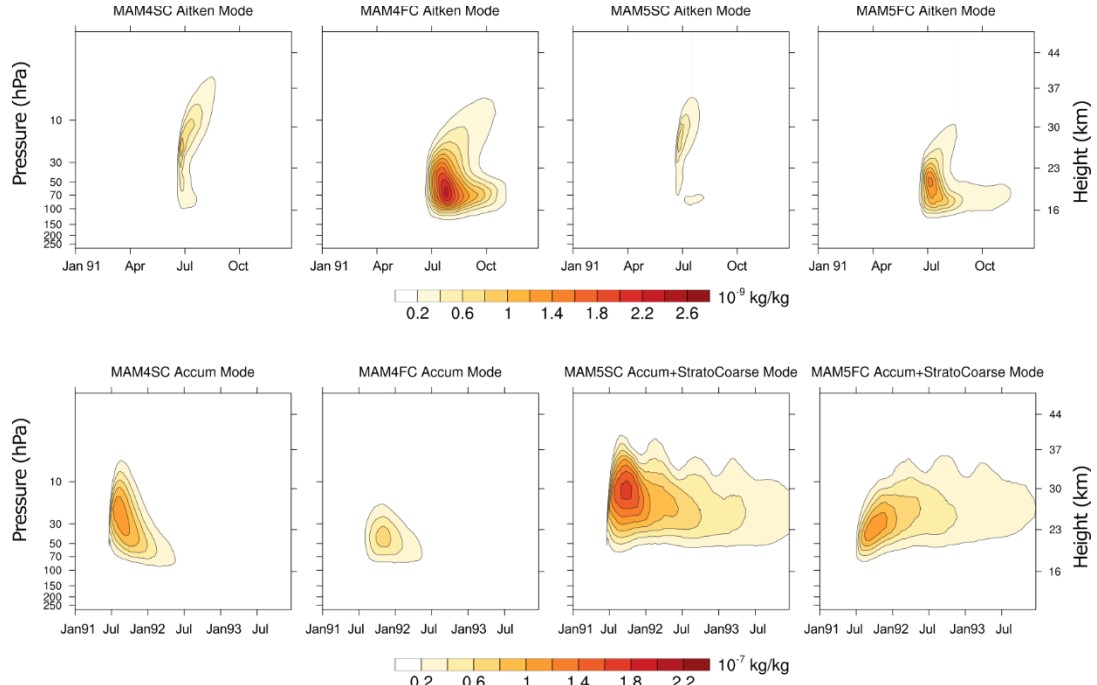

**Figure 3: Vertical profiles of tropical sulfate concentrations in the four experiments. The plots are divided by modes (Aitken mode in top panel with units of $10^{-9}$ kg/kg, accumulation mode or the sum of accumulation mode and stratospheric coarse mode for MAM5-PSA in bottom panel with units of $10^{-7}$ kg/kg). The plots for the Aitken mode aerosols span the year 1991 as the lifetime of the aerosols are within several months. The plots for the accumulation mode and stratospheric coarse mode aerosols span the years 1991 to 1993. The concentrations are averaged across longitude and latitude within the 30°S to 30°N band to remove interfering signals. Note that the top and bottom panels span different time periods.**

There are significant differences in concentrations and locations of sulfate between SC and FC. In the SC experiments, both the earlier Aitken mode sulfate and the later accumulation and stratospheric coarse mode sulfate remain at a higher altitude than the corresponding FC experiments. The highest concentrations occur between 10 and 30 hPa altitudes for SC, and between 30 and 70 hPa altitudes for FC, which may be explained by the higher $SO_2$ concentrations at 30-70 hPa in FC relative to SC, the smaller negative difference in OH concentrations at 30-70 hPa in FC relative to

SC, and the higher specific humidity at 30-70 hPa in FC compared to SC (Supplementary Figs. 1-4). For this same reason, the Aitken mode in FC generally has higher concentrations than SC after the eruption (up to $2.6\times10^{-9}$ kg/kg in FC compared to less than $1.0\times10^{-9}$ kg/kg in SC) in both the MAM4 and MAM5-PSA simulations. For the accumulation and stratospheric coarse modes, SC generally has higher concentration than FC several weeks after the eruption (up to $1.0\times10^{-7}$ kg/kg and $1.6\times10^{-7}$ kg/kg for MAM4SC and MAM5SC respectively compared to less than $1.0\times10^{-7}$ kg/kg in FC).

The main differences between MAM4 and MAM5-PSA are: (1) the inability of sulfate to progress past the accumulation mode in MAM4, and (2) the relatively short lifespan of the accumulation mode aerosols in MAM4 compared to the stratospheric coarse mode aerosols in MAM5-PSA. The shorter lifespan is a result of the MAM5-PSA's stratospheric coarse mode aerosols actually being smaller than MAM4's accumulation mode aerosols. Renaming from the accumulation mode to the stratospheric coarse mode in MAM5-PSA begins before the accumulation mode particles reach the maximum allowed size, which is easily reached in MAM4. The stratospheric coarse mode also has a reduced geometric standard deviation of 1.2 versus 1.6 for MAM4's accumulation mode, as described in Fig. 1. A smaller geometric standard deviation for stratospheric coarse mode in MAM5-PSA means that there are fewer super-coarse aerosols within the population. As super-coarse aerosols sediment more quickly, a smaller geometric standard deviation (1.2) leads to a longer lifespan for sulfate. In MAM4, constraining sulfate to the accumulation mode leads to aerosols, as well as the larger geometric standard deviation leads to a significant number of aerosols being super-coarse and quickly removed. Therefore, sulfate concentrations return to background level after roughly one year in MAM4 (as was also seen in E3SMv2-PA in Brown et al. (2024)) but this takes multiple years in MAM5-PSA.

### 3.2 Simulated burdens of stratospheric sulfate

Figure 4(a) shows the time series of the simulated stratospheric burden of sulfate between 80° S and 80° N, as well as the HIRS monthly observational data for sulfate burden for the same latitude bands taken from Baran and Foot (1994). Note that the burden derived from HIRS data assumes that the mean stratospheric temperature is 210K, which is increasingly inaccurate with more distance from the tropics. The derived sulfate burden from HIRS is calculated by taking the differences in the signal between the post-eruption period and an aerosol-free background, which means that it also includes tropospheric sulfate; however, the tropospheric sulfate burden from the Pinatubo eruption is relatively small compared to the stratospheric burden. Results from Ukhov et al. (2023) suggest that the eruption contributed about 0.03 to tropospheric AOD two weeks after the eruption, compared to about 0.13 to the stratospheric AOD, with tropospheric sulfate having a lifespan of roughly one month. Following the eruption of Mt. Pinatubo on June 15, 1991, sulfate burdens in all four model experiments rise rapidly, reach their peak value in November 1991, then begin to diminish over time. MAM4SC, MAM5SC and MAM5FC have about the same peak sulfate burden of about 5.5 Tg, occurring at roughly the same time in November 1991. The rate of increase for the sulfate burden, as well as the eventual peak sulfate burden, is the lowest in MAM4FC (about 4.8 Tg sulfur on November 30, 1991) due

to OH concentrations in the atmosphere being depleted by the extremely large amount of $SO_2$ produced by the eruption and being unable to replenish quickly. Starting in January 1992, the stratospheric burdens in MAM4 drop off more quickly compared to MAM5-PSA and MAM5FC agrees closely with HIRS until the end of 1992. In MAM4, sulfate in the accumulation mode grows until it reaches its maximum allowed geometric dry mean diameter of 0.48 microns (the Dg_hi value for the accumulation mode). Under such situations, the sulfate aerosol population increases in number (and therefore mass), but the size cannot increase any further. In MAM5-PSA, renaming is permitted, and the transfer from the accumulation mode to the stratospheric coarse mode begins when the geometric dry mean diameter is 0.40 microns (i.e. the Dg_lo value for the stratospheric coarse mode). Upon transfer to the stratospheric coarse mode, certain aerosol growth processes such as coagulation cease (as it is only allowed for the Aitken and accumulation modes in MAM), so the stratospheric coarse mode sulfates in MAM5-PSA tend to grow no larger in size, and the geometric dry mean diameter remains around 0.40 microns. In addition, the reduced geometric standard deviation of MAM5-PSA's stratospheric coarse mode (1.2) relative to MAM4's accumulation mode (1.6) also corresponds to a smaller proportion of super-coarse aerosols that are removed more quickly from the stratosphere, due to the settling velocity being roughly proportional to the second power of the aerosol diameter (Seinfeld and Pandis, 2016). These two reasons are responsible for the sulfate having a longer lifespan in MAM5-PSA than MAM4. The burden in MAM5-PSA continues to decrease almost linearly in the following years but does not return to background levels until 1995 (not pictured). For both MAM4 and MAM5-PSA, the full chemistry experiments have stratospheric sulfate burdens that decline more quickly after reaching peak values in November 1991 than the corresponding simple chemistry experiment.

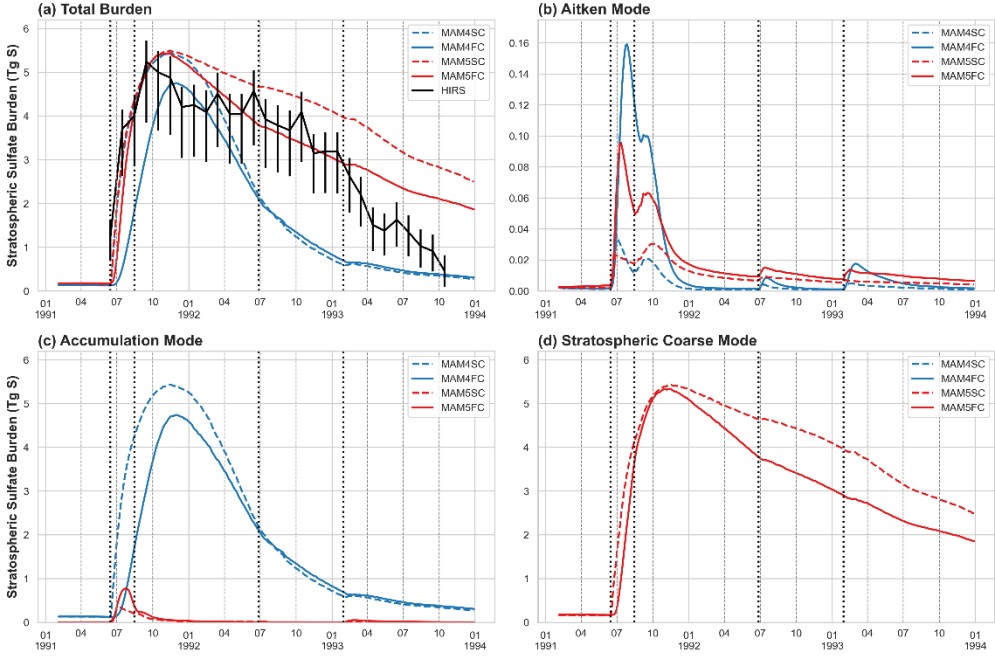

**Figure 4: (a) Comparisons of simulated stratospheric sulfate burdens (Tg S) between 80° S and 80° N from the four experiments with HIRS observations for years of 1991-1995. Black line depicts the estimates made from HIRS data in Baran and Foot (1994), with the black bars representing the margin of error. (b), (c), and (d) are similar comparisons for the**

**Aitken mode, accumulation mode and stratospheric coarse mode respectively (renaming from accumulation mode into the coarse mode does not exist in MAM4).**

Figures 4(b), 4(c) and 4(d) respectively show the variations in stratospheric sulfate burdens of Aitken mode, accumulation mode and stratospheric coarse mode. The results used in these plots are three-day averages on every third day. In MAM4, most of the sulfate burden exists in the form of the accumulation mode, as renaming into the coarse mode is not available. Around November 15, 1991 (the time of peak stratospheric sulfate burden), Aitken mode and accumulation mode contribute, respectively, 0.1% and 99.9% (0.3% and 99.7%) to total burden in MAM4SC (MAM4FC). For MAM5-PSA, the accumulation mode peaks briefly, but is soon almost entirely converted into the stratospheric coarse mode via renaming. Similar to the accumulation mode for MAM4, in both MAM5FC and MAM5SC the stratospheric coarse mode makes up more than 98% of the stratospheric sulfate burden on November 15, 1991, with about two-thirds of the remainder being in the accumulation mode and one-third in the Aitken mode.

It is worth noting that the peak values for the different aerosol modes do not occur concurrently. For the Aitken mode, all four experiments peak for the first time in late June and July (0.04 Tg S on June 30 in MAM4SC, 0.15 Tg S on July 30 in MAM4FC, 0.03 Tg S on June 30 in MAM5SC, and 0.1 Tg S on July 12 in MAM5FC). The results here agree with those in Fig. 2, where aerosol formation is faster in SC compared to FC, and MAM5FC is faster than MAM4FC due to replenishment of OH radicals. Several months after the Pinatubo eruption, there is again an increase in Aitken mode burden due to the eruption of Mount Hudson between August and October. The Mount Spurr and Lascar eruptions of 1992 and 1993 respectively are also visible here.

For the accumulation mode, the peak burden of 5.5 Tg S (4.8 Tg S) occurs on November 15 (November 30) in MAM4SC (MAM4FC). Much like for Aitken mode, the peak value for MAM4FC occurs later than MAM4SC as a result of the slower start in terms of sulfate formation. For MAM5-PSA, accumulation mode stratospheric sulfates are mostly transitory in nature, quickly renaming into the stratospheric coarse mode. The accumulation mode of MAM5SC and MAM5FC has small peak values of 0.4-0.8 Tg S during July 9-24. The stratospheric coarse mode burdens peak around November 15 for both MAM5-PSA experiments at around 5.5 Tg S.

Figure 5 shows the stratospheric burdens of sulfate for different modes and latitudinal regions of 30 °N-80 °N, 30 °S-30 °N, and 80 °S-30 °S. As with Fig. 4, for all regions the vast majority of the stratospheric sulfate burden exists in the accumulation mode for MAM4 and in the stratospheric coarse mode for MAM5-PSA. It should be noted that the total stratospheric sulfate burden exhibits different changes with time for MAM5-PSA. Between 30 °S and 30 °N, the total stratospheric burden in all experiments declines the fastest compared to the other two regions due to both gravitational settling of sulfate and its poleward transport due to the Brewer-Dobson circulation. Between 30 °N-80 °N, the stratospheric sulfate burden peaks in February and March of 1992, as opposed to October 1991 in 30 °S-30 °N, due to the time required for aerosol transport northward via the Brewer-Dobson circulation. The stratospheric sulfate burden has a second peak in January 1993 at 30 °S-30 °N and 30 °S-80 °S corresponding to the Lascar eruption,

though this does not counteract the overall decline over time. Between 30 °S-80 °S, the stratospheric sulfate burden peaks in November 1991 for both MAM5-PSA experiments, and afterwards remains relatively steady. For MAM5-PSA the stratospheric sulfate burden appears to oscillate annually in the southern hemisphere, with high values around October-November and low values around April of 1992 and 1993.

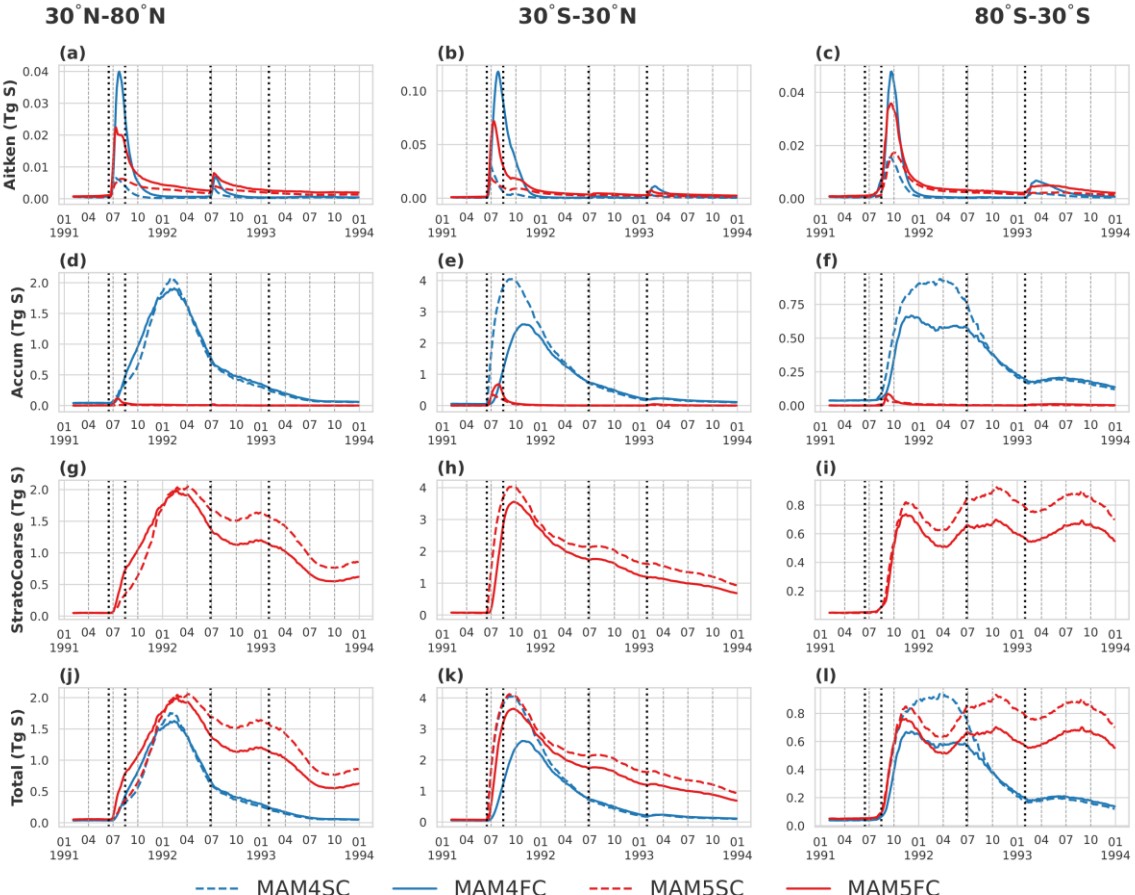

**Figure 5: Simulated stratospheric sulfate burden (in Tg S) divided by mode and latitudinal region. The vertical lines represent the Pinatubo, Cerro Hudson, Spurr and Lascar eruptions respectively. Note the differing vertical axes in different plots.**

It can also be seen in Fig. 5 that the relative latitudinal contribution to the total stratospheric sulfate burden changes over time. The burden between 30 °S and 30 °N constitutes about two-thirds or more of the stratospheric sulfate burden in the months after the Pinatubo eruption. Starting from April of 1992, the stratospheric burden of 1.7-2.1 Tg S between 30 °N-80 °N is similar in magnitude to the stratospheric burden between 30 °S and 30 °N in all four experiments. Starting from 1993, the stratospheric burden south of 30 °S in MAM5-PSA begins to make up a much larger portion of the total between 80 °S and 80 °N (about 0.8 Tg S out of the total of about 4.1 Tg S), due to the stratospheric coarse mode burdens not decreasing. This accounts for the slower decline of the total stratospheric burden in MAM5-PSA as seen in Fig. 4(a).

Since the independent stratospheric coarse mode is newly added in this study, Figure 6 depicts the distribution of MAM5-PSA aerosol volume particle size distribution, as compared with WOPC observations (Deshler et al., 1993; Deshler et al., 2003). MAM4 has been omitted as it does not include coarse mode sulfate. It can be seen that with the exception of December 30, 1991, MAM5FC generally better matches WOPC observations in terms of coarse mode mass than MAM5SC. However, MAM5FC coarse mode size is biased towards being smaller than the observations. The relatively smaller coarse mode aerosols may also help to explain the excessively long lifespan later seen past 1992 in Figs. 4 and 5.

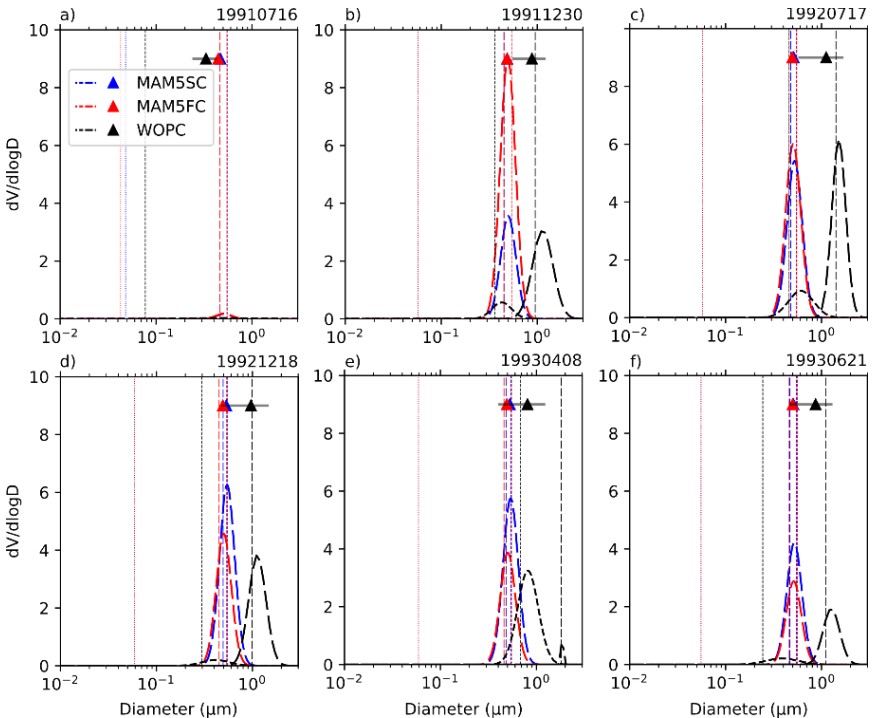

**Figure 6: Comparisons of stratospheric aerosol size distributions from MAM5SC and MAM5FC with observations from WOPC for 1991–1993. WOPC launches are samples taken from the 18 km measurements and matched to the nearest model height and grid cell over Laramie, Wyoming (41.3° N, 105° W). The vertical dotted, dashed and long-dashed lines represent the geometric mean diameters of the Aitken, accumulation and stratospheric coarse modes respectively. The triangles represent the effective diameters calculated from the size distribution.**

### 3.3 Process analyses

To better understand the differences between the experiments and how the aerosols transition between the different modes, the tendencies (i.e., the rate of mass change into or out of a certain mode) between 80 °S and 80 °N are plotted in Fig. 7. The left column represents the tendencies over time, while the right column represents the cumulative tendency, integrated over time. The tendencies are integrated over the three dimensions of: all longitudes, the above-mentioned latitude range, and vertically above the tropopause. Here it is assumed that these rates apply only to sulfate and other aerosol contributions are negligible within the stratosphere. As described in the methodology section 2.1,

nucleation, coagulation and renaming represent mass exchange between the different aerosol modes. Condensation represents the mass growth within the same aerosol mode.

With respect to nucleation, Figure 7(a) shows the magnitude of the tendency for the formation of Aitken mode aerosols through the nucleation of gaseous $H_2SO_4$ (NUCL) is generally higher in the FC experiments compared to the SC experiments, which leads to the higher Aitken mode sulfate concentrations in FC than in SC as shown in Fig. 4(b). Although one would expect SC to have faster nucleation without the bottleneck of OH concentrations, new particle

formation by nucleation competes against aerosol growth by condensation. Condensation is stronger in the SC experiments than FC for the first five months after the eruption, predominantly in the accumulation mode condensation for MAM4 (Figs. 8 (c)(d)) and in the stratospheric coarse mode for MAM5-PSA (Figs. 8 (e)(f)). This leads to competition with the nucleation process, so FC has stronger nucleation than SC. Due to the higher OH concentrations in the SC experiments, the nucleation process begins and ends earlier in the SC experiments compared to FC.

Considering the cumulative mass changes depicted in Fig. 7(b), the curves for the SC experiments flatten out (i.e., the microphysical process in SC is mostly concluded) earlier than those for FC.

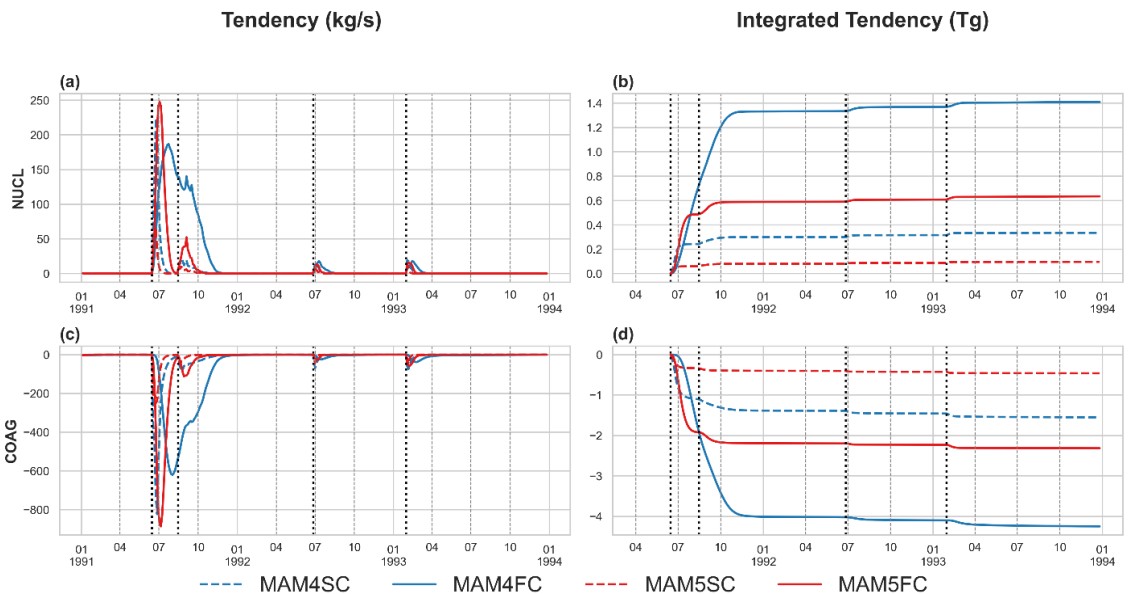

**Figure 7, part 1: A series of figures for the relevant tendencies for each mode in each experiment in the stratosphere between**
**80°S and 80°N., in terms of aerosol mass. See Figure 1(a) for description of processes. Positive values represent gained mass from the associated microphysical process from the perspective of the aerosol mode in question. NUCL represents the aerosol mass gain for Aitken mode due to nucleation (i.e., aerosol formation), COAG represents Aitken mode mass loss due to coagulation into the accumulation mode. The left column plots represent the tendency values over time, while the right column represents the cumulative mass change due to the associated microphysical process (i.e. tendency integrated over**
**time). The vertical lines represent the Pinatubo, Cerro Hudson, Spurr and Lascar eruptions respectively. Note the differing vertical axes in different plots.**

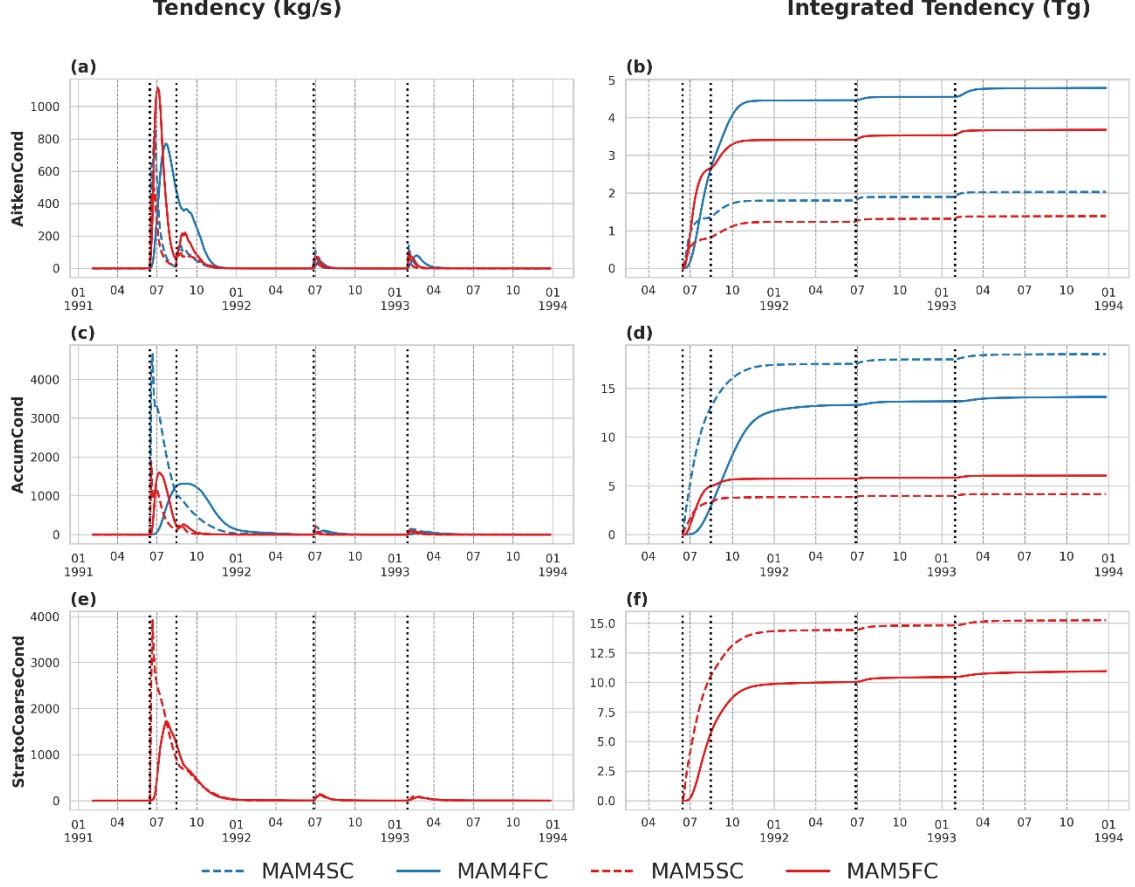

**Figure 8: Tendency plots for condensation in the different aerosol modes, similar to those for nucleation and coagulation in Fig. 7. Positive values represent mass transition from the gaseous phase to the aerosol. Note the differing vertical axes in different plots.**

With respect to coagulation, Figs. 7(c)(d) show the aerosol mass loss rates (negative values) of the Aitken mode due to coagulation into the accumulation mode (COAG). The magnitude of the coagulation generally follows that of the nucleation process, as the strength of the coagulation process is proportional to Aitken mode aerosol number and mass.

With respect to the condensation processes, Figs. 8(a)-(f) show the condensation processes for each of the three modes. Fig. S6 in the supplementary shows the sum of condensation processes across the three modes for reference. Across the first five months after the eruption, for Aitken mode condensation, FC has stronger condensation tendencies than SC corresponding to the larger burden of Aitken mode aerosols seen in Figure 4(b). For the accumulation mode, condensation is stronger for MAM4 than MAM5-PSA likewise corresponding to the very small accumulation mode burden in MAM5-PSA (Fig. 4(c)). For the stratospheric coarse mode, MAM5SC has a much stronger condensation tendency than MAM5FC corresponding to the former having a higher stratospheric coarse mode burden (Fig. 4(d)). It should be noted that MAM4SC similarly has stronger condensation tendency for the accumulation mode than

MAM4FC, as the sulfate does not grow beyond the accumulation mode. After five months, condensation concludes in all of the experiments except for MAM4FC, due to delays from its OH bottleneck.

Figs. 9(a)(b) show the aerosol mass loss rate of the Aitken mode due to renaming into the accumulation mode (RNMaa). Similar to coagulation above, the strength of RNMaa is also dependent on the preceding processes of nucleation and condensation, which provide the large Aitken mode particles necessary for renaming. Since nucleation and Aitken mode condensation is stronger in FC than the SC experiments, this is also true for RNMaa.

Figs. 9(c)(d) represent the aerosol mass gain rate of the stratospheric coarse mode due to renaming from the accumulation mode (RNMasc). Similar to RNMaa, the relative strength is determined by the prior steps, here being RNMaa, COAG and AccumCond, leading to MAM5FC being stronger than MAM5SC.

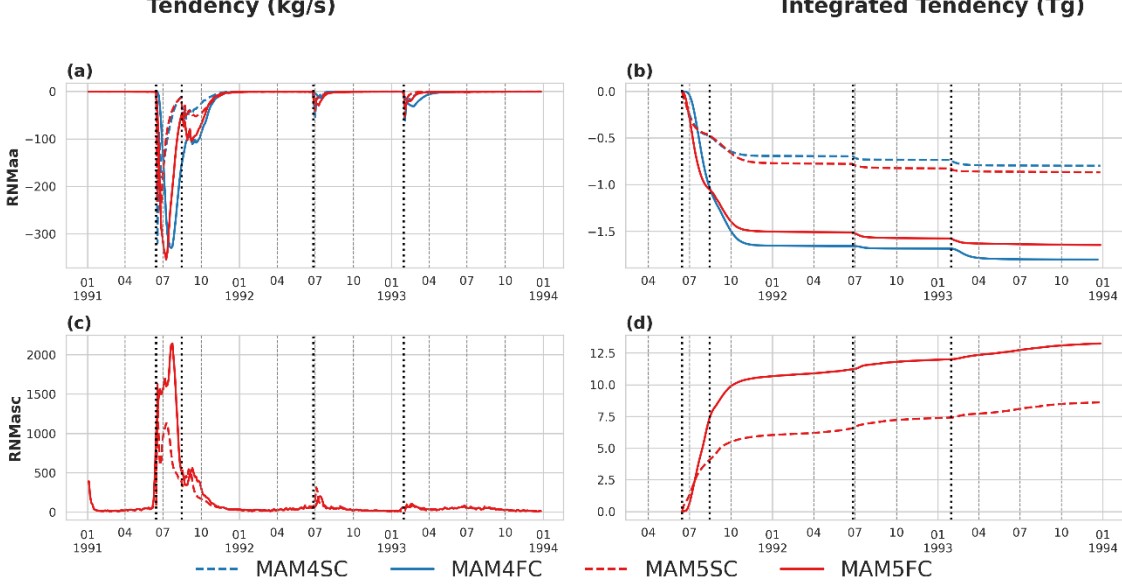

**Figure 9: Tendency plots for renaming, i.e. the transition of mass between aerosol modes due to the aerosol population reaching a prerequisite size, as described in Figure 1(a). RNMaa represents the mass transition from the Aitken mode to the accumulation mode (renaming is only possible in this direction, and cannot occur in the reverse), and the negative value represents the mass loss of the Aitken mode. RNMasc represents the mass transition from the accumulation mode to the stratospheric coarse mode (also only possible in this direction), and the positive value represents the mass gain of the stratospheric coarse mode. Note the differing vertical axes in different plots.**

Overall, SC has stronger condensation processes (sum for all modes) and weaker nucleation (i.e. initial formation of Aitken mode sulfates) relative to FC over the first five months, again showing the competition between condensation and nucleation for $H_2SO_4$. Because coagulation and renaming are dependent on nucleation and condensation, FC is stronger than SC for coagulation and renaming. With respect to the differences in processes between MAM4 and MAM5-PSA, MAM4 has stronger nucleation, coagulation and renaming processes due to weaker competition from condensation.

We have also examined the microphysical processes divided by latitudinal regions (Supplementary Figs. 7 and 8). The same patterns as above apply between 30 °S and 30 °N. Above 30 °N and below 30 °S the same signal from the Pinatubo is still present, though slightly delayed due to the time that it took for the aerosol to transport poleward. Signals from other eruptions (Spurr and Lascar) are also present. The main difference is in RNMasc. Above 30 °N

and below 30 °S, renaming into the stratospheric coarse mode for MAM5-PSA continues long after the Pinatubo eruption, contributing to the long lifespan of stratospheric sulfate in MAM5-PSA as depicted in Figures 4 and 5. This did not occur between 30 °N and 30 °S, where the stratospheric burden also declines more rapidly as seen in Fig. 5. It is also worth noting that the same pattern does not exist for the other tendencies, and only RNMasc continues to a significant degree past 1992. The condensation tendencies are also not evenly divided across different latitudinal

regions, with 30 °S-80 °S and 30 °N-80 °N having stronger condensation tendencies for MAM4FC in the Aitken and accumulation modes, and for MAM5FC in the Aitken and stratospheric coarse modes. Notable increases in condensation can be seen during the Spurr and Lascar eruptions, though it may also be due to Aitken mode aerosols from the original Pinatubo eruption being transported away from the equator and continuing their growth elsewhere.

**3.4 Simulated AOD and net solar flux**

Figure 10 depicts the anomaly of simulated AOD from the four experiments and AVHRR observations from 1991-1993, relative to the corresponding data from 1989 (i.e., the monthly averages during 1991-1993 subtracted by the corresponding monthly average in 1989 for the same pixel), and masked according to the observed area by AVHRR. The AVHRR AOD anomaly had a maximum value in the range of 0.4-0.45 between 10 °N and 10 °S starting from

July 1991, corresponding to the Pinatubo eruption. The four simulations do not have as high of an AOD anomaly for the Pinatubo eruption, with an anomaly of 0.25-0.3 for MAM4SC, 0.15-0.2 for MAM4FC, and 0.35-0.4 for both MAM5SC and MAM5FC experiments. In this respect, MAM5-PSA better captures the AOD anomaly due to the Pinatubo eruption. The simulated AOD maximums occurs slightly north of the equator, to the north of the AOD maximum in the AVHRR observation.

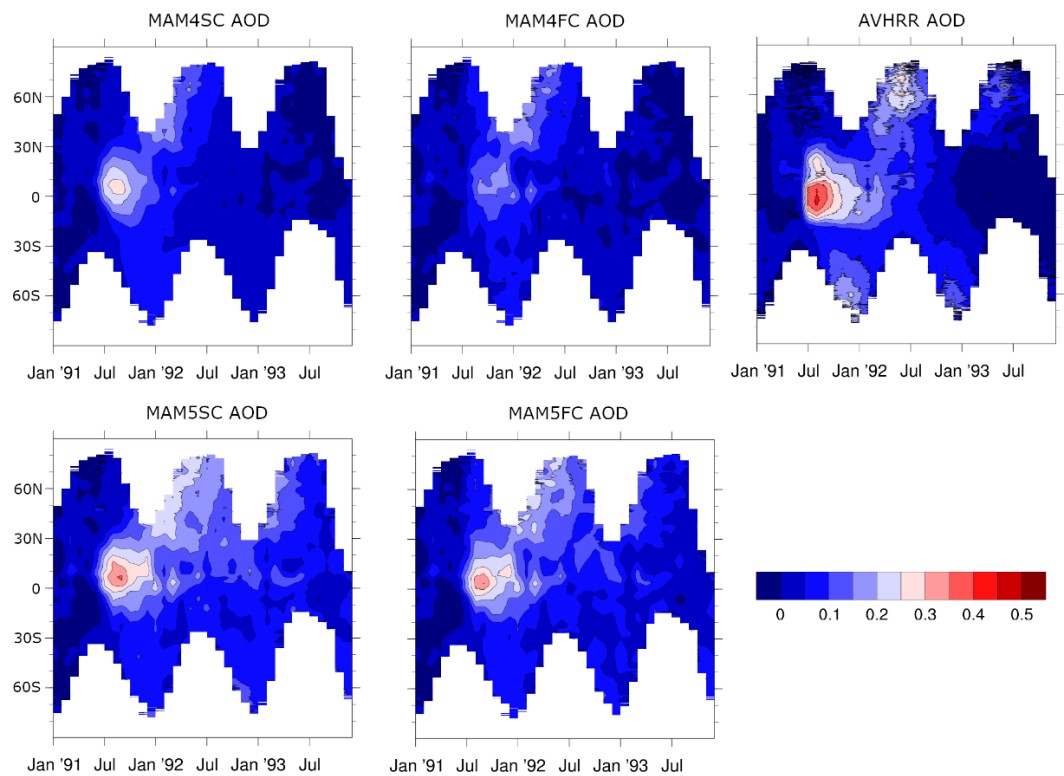

**Figure 10: Latitude vs. time plots of model simulated AOD and AVHRR observed AOD anomaly values relative to 1989 AOD output or observations respectively, to eliminate contributions from the troposphere and background stratospheric AOD. Model output was masked to correspond to areas that AVHRR had observation data for. Results were averaged across longitude.**

The four experiments capture well the northward transport of the AOD anomaly in the latter half of 1991 and the first half of 1992, also matching the generally higher stratospheric sulfate burden above 30 °N compared to below 30 °S (Fig. 5). In all four experiments and the AVHRR observations, the AOD anomaly diminished over time following the Pinatubo eruption, returning towards 0 starting from 1993. This decay is weaker in the MAM5-PSA experiments compared to MAM4 and the AVHRR results, with lingering positive AOD anomalies of 0.05-0.1 throughout 1993, matching the longer lifespan of the stratospheric sulfate in MAM5-PSA as seen in Fig. 4.

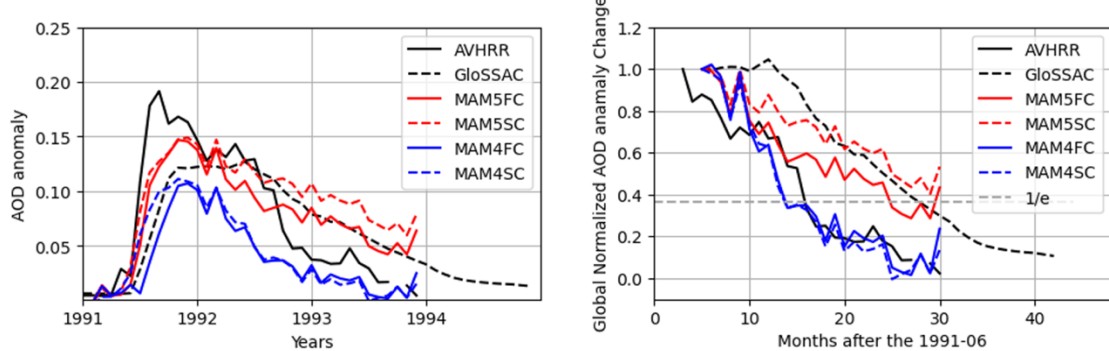

**Figure 11: Comparison of AOD anomaly between simulations and observations. The left panel shows the time evolution of monthly AOD anomaly values from simulations and AVHRR and GloSSAC observations, while the right panel shows the time evolution of normalized AOD anomaly.**

We have evaluated the characteristics of the time evolution of simulated global AOD anomaly and its e-folding time against observational data. The time evolution of global monthly AOD anomaly, averaged between 60°S–60°N, is depicted in the left panel of Fig. 11. The AVHRR and the Global Space-based Stratospheric Aerosol Climatology (GloSSAC) were two satellite-based AOD observations that provided global coverage during the Mt. Pinatubo eruption. GloSSAC offered zonal values with a latitudinal resolution of 5° and uniform spatio-temporal coverage up

to the year 1994. However, due to instrument saturation for optical depths around 0.15, GloSSAC was less accurate in the center of tropical clouds during the first months after the Mt. Pinatubo eruption. It is also worth noting that GloSSAC data includes only stratospheric AOD, and is missing tropospheric AOD anomalies that are included in AVHRR and the model results. Conversely, AVHRR could only measure stratospheric AOD values larger than 0.01, making it less reliable when AOD values were low (Russell et al., 1996; Quaglia et al., 2023). To calculate AOD

anomaly values, background levels (from June 1989 to May 1991) were removed from both observations and simulations. The MAM5FC and MAM5SC simulations produced a reasonable global mean AOD anomaly peak of around 0.15 in November 1991, when the simulated sulfate aerosol burden peaked (Fig. 4). These peaks were higher than the GloSSAC peak (0.12) but lower than the AVHRR peak (0.19). The MAM4FC and MAM4SC simulations produced AOD anomaly peaks (0.11) that were smaller than GloSSAC, indicating an apparent underestimation of

AOD anomaly strength. The MAMFC simulation showed a faster AOD anomaly decay than MAM5SC, with values aligning closely with GloSSAC in 1993. In contrast, the MAM4FC and MAM4SC simulations consistently underestimated AOD anomaly compared to GloSSAC from October 1991 to December 1993.

       The e-folding time, calculated as the time between the maximum and the 1/e value, was 13 months in AVHRR and

16 months in GloSSAC. The difference between the two measurements is due to the nature of these instruments. The simulated e-folding time in MAM4FC and MAM4SC was 10 months, while MAM5FC and MAM5SC had e-folding times of 19 months and 25 months, respectively.

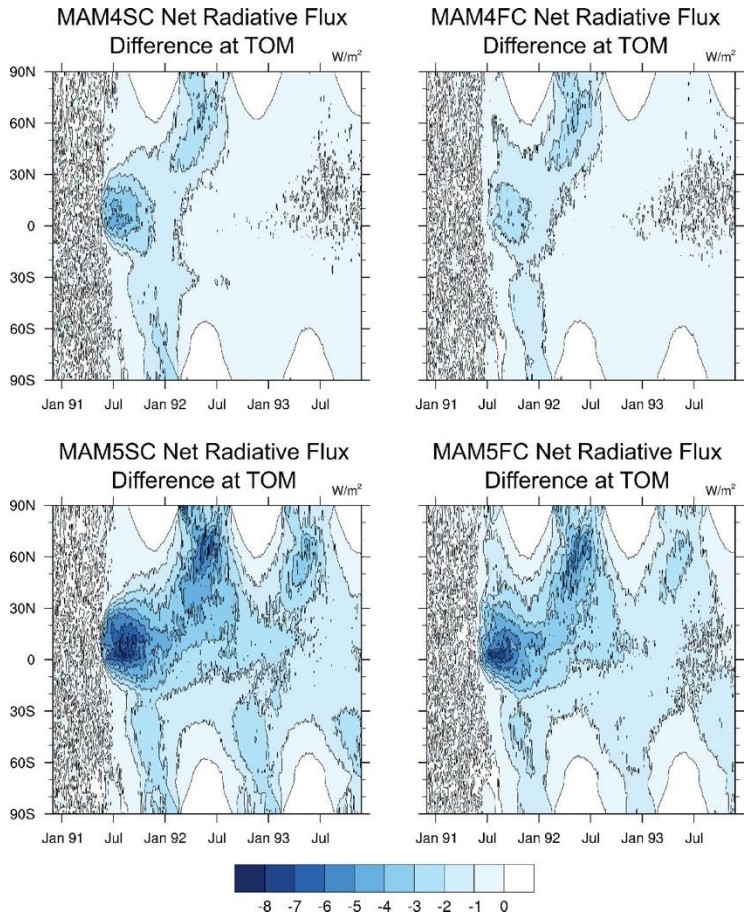

**Figure 12: Latitude vs. time plots of the top-of-model net solar flux difference between the four experiments and corresponding model simulations with emissions from volcanic eruptions shut off. Units in W/m².**

Figure 12 depicts the differences in the top-of-model net solar flux in the four experiments with and without the emissions from volcanic eruptions (negative values representing cooling from the volcanic aerosol). The location of the maximum of the net solar flux difference agrees well with the location of the AOD anomalies in Fig. 10. The maximum of the net solar flux is simulated to be about -6 W/m² for MAM4SC, -4 W/m² for MAM4FC, and -10 W/m² for both MAM5-PSA experiments. Considering the more accurate AOD simulation in MAM5-PSA, it is expected that MAM5-PSA also better captures climate effects from the Pinatubo eruption.

The time variations of the global mean top-of-model net (shortwave plus longwave) radiative flux anomaly shows that, during October-November of 1991, MAM4 experiments produce weaker peak values of around -1.5 W/m² while MAM5-PSA produces peak values of roughly -3.0 W/m² (Fig. 13). The weaker cooling in FC corresponds to the smaller stratospheric sulfate burden as seen in Fig. 4, while SC's stronger cooling also corresponds to its larger stratospheric sulfate burden. Similar to the sulfate burden as discussed in section 3.2, the tropospheric AOD from the Pinatubo eruption is not significant compared to the stratospheric values. The MAM5-PSA results agree closely with the those simulated by CESM experiments in Mills et al. (2017) as well as the E3SMv2-SPA results from Brown et

al. (2024), where the peak values are also roughly -3.0 W/m$^2$. It also agrees well with the simulation results from Hansen et al. (2005) up to early 1992, which also reports a peak value of about -3.0 W/m$^2$ across their five runs, though the net radiative flux signal is slower to diminish around July 1992, concurring with the stratospheric sulfate burden depleting too slowly in MAM5FC. In general, the net radiative flux differences closely follow the same patterns as the stratospheric sulfate burden depicted in Fig. 4, suggesting that differences in chemistry between SC and FC, and the additional stratospheric coarse mode in MAM5-PSA, first influence stratospheric sulfate burdens, which in turn affect the net radiative flux.

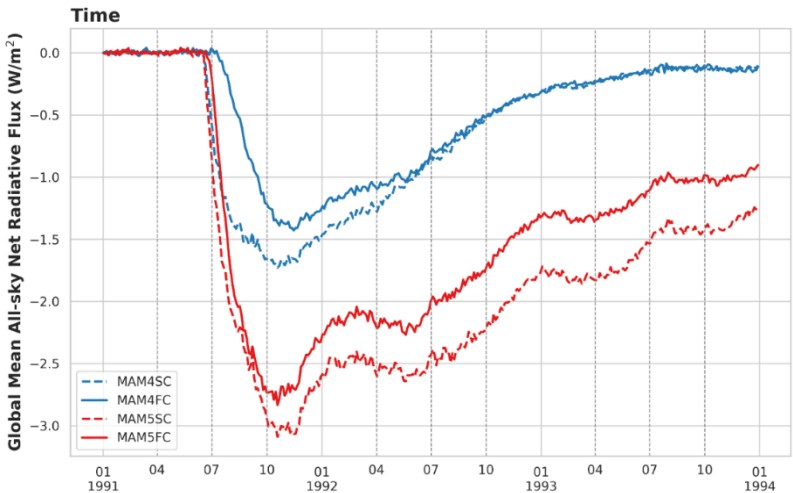

**Figure 13: Global mean net radiative flux anomaly in the four experiments, where negative values represent a net upward flux. The anomaly is the difference with the corresponding model experiment with the volcanic emissions shut off.**

## 4 Conclusions and discussions

In this work, we have implemented a fifth stratospheric coarse mode (MAM5-PSA) on top of the original four-mode version Modal Aerosol Module (MAM4) used in E3SMv2. We also consider a more complex "full chemistry (FC)" set-up, which includes the addition of a series of chemical reactions which allow the simulation of OH radical replenishment compared to "simple chemistry (SC)" with prescribed OH concentrations. We have carried out a series of simulations of the Pinatubo eruption for the time period of 1991-1993 to better understand the aerosol microphysical processes including nucleation, coagulation and condensation. We have also compared model simulation results with measurements of stratospheric sulfate burden, aerosol size, AOD and net radiative flux.

Compared to SC, FC leads to large differences in both the temporal variations and the vertical distributions of sulfate concentrations. Directly after the eruption, the SC experiments had the fastest increase in sulfate concentrations, as FC was limited by OH required to oxidize $SO_2$. The increase in MAM5FC was significantly faster than MAM4FC however due to the OH radical replenishment. Vertically, sulfate distributions were generally at lower altitudes in FC

compared to SC, caused by the differences in $SO_2$, OH, and specific humidity within the stratosphere. Starting in 1993, stratospheric sulfate burdens in all four of the experiments began to diminish, but the process was slower in MAM5-PSA because of the stronger sedimentation in MAM4 relative to MAM5-PSA. The slow removal of stratospheric sulfate burden in MAM5-PSA occurs mostly in the high latitudes but is faster near the equator.

Model evaluations indicate that the simulation with both MAM5-PSA and full chemistry (i.e. MAM5FC) can better capture observations such as sulfate burden, aerosol size, AOD and net radiative flux. With respect to the stratospheric sulfate burden, MAM5FC (compared to MAM4SC, MAM4FC and MAM5SC) agreed best with the HIRS observations from the eruption of the volcano to the end of 1992. Compared to WOPC observations of aerosol size, MAM5FC better captures observations and has comparable mass in the coarse mode. We also assessed the time evolution and e-folding time of simulated global AOD anomaly against satellite observations (AVHRR and GloSSAC). The MAM5FC and MAM5SC simulations generally produced reasonable AOD peaks and decay patterns, while MAM4FC and MAM4SC tended to underestimate AOD strength and overestimate the decay rate. From July 1991 to the end of 1992, simulated net radiative flux anomalies are in the range of -2.8 to -1.3 $W/m^2$, agreeing well with observed net radiative flux anomalies within the range of -3.8 to -1 $W/m^2$ (Brown et al., 2024).

We further analyzed the tendencies associated with the microphysical processes related to the growth of stratospheric sulfate: nucleation (NUCL), coagulation (COAG), condensation (AitkenCond, AccumCond, StratoCoarseCond), and renaming (RNMaa and RNMasc). Due to the limited amount of $H_2SO_4$ in the stratosphere, the nucleation and condensation processes compete over $H_2SO_4$ until condensation ceases about five months after the eruption. Overall, over these five months, FC has weaker condensation processes (sum for all modes) and consequently stronger nucleation, coagulation and renaming relative to SC, since coagulation and renaming are dependent on nucleation and condensation.

The usage of the Modal Aerosol Module in this study allows us to more accurately reflect the aerosol growth process over time. For the purpose of simulating volcanic eruptions, this is an improvement over bulk schemes which cannot represent aerosol size (Gao et al., 2023) and allows for more accurate simulation of aerosol lifespan. Similar to (Brown et al., 2024) and (Mills et al., 2016), issues with the excessively short lifespan of stratospheric sulfates in default MAM (i.e. MAM4 in this study) are addressed with modifications of aerosol size parameters to better reflect coarse mode stratospheric sulfates. However, instead of overwriting the original coarse mode properties and interfering with the simulation of other coarse mode aerosols like sea salt and dust, we separate stratospheric sulfate into its own mode. This not only prevents interference with the other coarse mode aerosols, but also allows us to investigate the aerosol microphysical processes of sulfates in particular.

A number of factors contribute to the uncertainty of our results. One important factor is the known dry bias in E3SM's stratosphere (Christiane Jablonowski, personal communication, 2024), since water vapor plays an important role in the $HO_x$ cycle relevant to the initial sulfate formation, as well as later sulfate growth and sedimentation. Another issue

is the presence of minor volcanic eruptions such as Cerro Hudson, Spurr and Lascar, which produced much less $SO_2$ than Pinatubo, but still significant enough to leave a distinguishable signal following Pinatubo's eruption. A third factor is the interaction between the Pinatubo eruption and stratospheric ozone. Volcanic eruptions are known to affect stratospheric ozone concentrations (Peng et al., 2023) directly and indirectly through factors such as the decreases in tropospheric temperatures, modified circulation and the changes in the stratospheric chemical composition (Dhomse et al., 2015). The capabilities of E3SM in simulating stratospheric ozone in the unique circumstances of a volcanic eruption are not well studied. Further understanding of these factors would greatly aid in the accurate simulation of Pinatubo and other volcanic eruptions.

In summary, our study shows that a physical representation of aerosol size distribution while also considering the impact of interactive OH concentrations can better capture observations of stratospheric sulfate burden, aerosol size, AOD and net radiative flux, which have important implications for future studies of stratospheric aerosols.

**Code and data availability.**

AVHRR observational data was downloaded from https://www.ncei.noaa.gov/data/avhrr-aerosol-optical-thickness/access. The VolcanEESM emissions files used in this study can be found at https://svn-ccsm-inputdata.cgd.ucar.edu/trunk/inputdata/atm/cam/chem/stratvolc/. The source code and run scripts used in this study can be found at https://zenodo.org/doi/10.5281/zenodo.12734295.

**Author contributions.**

Conceptualization: AH, XL and ZK. Data curation: AH, ZK and BW. Formal analysis: AH. Funding acquisition: XL, DB and KP. Investigation: AH. Methodology: AH, XL and ZK. Project administration: XL, HW, SX, DB and KP. Resources: XL, DB and KP. Software: AH implemented MAM5-PSA into E3SMv2 with assistance from ZK, MW, BW and ZL. Original MAM5-PSA implementation in E3SMv1 by ZK. ZK, MW, HW and QT enabled and improved the chemistry scheme that "full chemistry" is based on. Supervision: XL, DB and KP. Validation: AH. Visualization: AH. Writing – original draft preparation: AH. Writing – review and editing: all authors.

**Competing interests.**

Some authors are members of the editorial board of Atmospheric Chemistry and Physics.

**Disclaimer.**

**Acknowledgements.**

XL, AH, BW, HB, DB, and KP acknowledge the support of this work by the Laboratory Directed Research and Development program at Sandia National Laboratories, a multimission laboratory managed and operated by National Technology and Engineering Solutions of Sandia, LLC., a wholly owned subsidiary of Honeywell International, Inc., for the U.S. Department of Energy's National Nuclear Security Administration under contract DE-NA-0003525. XL, ZK, ZL, MW, HW, QT, and SX acknowledge the support by the U.S. Department of Energy (DOE), Office of Science, Office of Biological and Environmental Research, Earth and Environmental System Modeling program as part of the Energy Exascale Earth System Model (E3SM) project. The work at the Lawrence Livermore National Laboratory (LLNL) was performed under the auspices of the U.S. DOE by LLNL under Contract DE-AC52-07NA27344. The Pacific Northwest National Laboratory (PNNL) is operated for DOE by Battelle Memorial Institute under contract DE-AC05-76RLO1830.

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
