# Peer review of "Size-resolved process understanding of stratospheric sulfate aerosol following the Pinatubo eruption"

_EGUsphere, 2024_

## Author Response (AR3)

Response to the handling editor:

Thank you very much for taking the time to update the questions and comments to the latest version of the manuscript. Please find our responses below:

1. L251ff: What is the difference between R1 and R2? Why is R1 included?
   **Response:**
   The listed reactions are all of the reactions within the default model's chemistry processor package. R1 and R2 are listed in Table 3 of Emmons et al. (2010) as part of the MOZART chemistry scheme which "simple chemistry" is based on. Within the model, these two reactions have different reaction rates in the absence and presence of a reaction chaperone M, respectively. To clarify, we have added the following sentence before the list or reactions: "R1 and R2 have different reaction rates, with the latter requiring an additional chaperone molecule."

   | | | | |
   |---|---|---|---|
   | DMS + OH | → | SO2 | $9.60E\text{-}12 \cdot \exp(-234/T)$ |
   | DMS + OH | → | .5·SO2 + .5·HO2 | $1.7E\text{-}42 \cdot \exp(7810/T) \cdot [M] \cdot 0.21 / (1 + 5.5E\text{-}31 \cdot \exp(7460/T) \cdot [M] \cdot 0.21)$ |

2. L254ff: It seems intermediate steps are left out in this chemistry? Or why don't the equations balance, e.g. R4, R6. R7?
   **Response:**
   Intermediate steps are left out to simplify calculations and reduce computational costs. Some byproducts (e.g., $O_2$, $H_2O$) are not produced in sufficient quantities to significantly influence their concentration in the atmosphere (e.g., $O_2$ production in R4) or affect their participation in $SO_2$ production or other reactions, so these byproducts are simply omitted from the reactions. This is why some of the equations do not balance.

3. Fig 5 caption: please include a warning of different scales on vertical axes
   **Response:**
   Thank you for your comment. A note has been added to the end of the Fig. 5 caption: "Note the differing vertical axes in different plots."

4. L481: "In order" The authors use this throw away phrase too much. It is almost never necessary.
   **Response:**
   Thank you for the suggestion. Several instances of the phrase have been removed. Now it is "To examine the differences…" in Section 2.3 and "To better understand…" in Section 3.3.

5. Figs 7,8, and 9, please include warnings of different scales on vertical axes
   **Response:**
   Thank you for your comment. Warnings have been added to the end of the captions of Figs. 7,8 and 9: "Note the differing vertical axes in different plots."

6. Fig 13: It would be worth pointing out again here why the SC models estimate more cooling in both MAM4/5 than the FC models.
   **Response:**

We have revised the sentence to explain the difference in cooling between SC and FC in MAM4/5:

"The time variations of the global mean top-of-model net radiative flux shows that, during October-November of 1991, MAM4 experiments produce weaker peak values of around -1.5 W/m$^2$ while MAM5-PSA produces peak values of roughly -3.0 W/m$^2$ (Fig. 13). The weaker cooling in FC corresponds to the smaller stratospheric sulfate burden as seen in Fig. 4, while SC's stronger cooling also corresponds to its larger stratospheric sulfate burden."

7. L208: this text and the bullet list that follows refers to "processes", but the list includes both physical processes and the technical/computational process of renaming. This writing style can be confusing to readers who may not be familiar with the material. Please reorganize slightly (i.e., remove renaming from the list of physical processes) and be more explicit about the difference between the models' representation of physical processes versus technical steps.
**Response:**
Thank you for pointing this out. To clarify this, we have removed renaming from the list of physical processes. We've also added the following sentences to avoid confusion for the readers:
"Each of these processes have their physical rules within MAM, and represent an actual physical process with the exception of renaming, which is a technical step in MAM."

"We note that renaming is a computational step in MAM, and does not correspond to a physical process".

8. L232: symbols used in the figure should be defined in the figure caption.
**Response:**
We have restored the definition of Dg, Dg_hi and Dg_lo in the Fig. 1 caption:
"Dg, Dg_hi, and Dg_lo represent the initial values of the geometric dry mean diameter and its upper and lower limits, respectively."

9. L282: Is this justification for the smaller SO2 emission for Pinatubo compared to observations something that is established, or more speculated? There is also evidence that the stratospheric injection by Pinatubo is actually much smaller than the total observed emission, see Ukhov et al. (2023).
**Response:**
We used emissions of 10 Tg following Mills et al. (2016). Ukhov et al. (2023) specifically mentions this assumption used in Mills et al. (2016) and agrees with it.
We have added a brief justification:
"Pinatubo eruptions are assumed to have occurred on June 15 1991, with 10 Tg of SO$_2$ evenly emitted between 18 and 20 km altitude. This corresponds to the mass detected in the stratosphere by the TIROS Operational Vertical Sounder and Total Ozone Mapping Spectrometer 7-9 days after the beginning of the eruption (Guo et al., 2004)."

10. L330: please point out here and in figure caption the figure displays tropical sulfate concentrations
**Response:**

Thank you for the suggestion. We have pointed this out at the start of the paragraph and the caption.
"Figure 3 depicts the vertical profiles of tropical sulfate concentrations"
"Figure 3: Vertical profiles of tropical sulfate concentrations in the four experiments."

11. L366: "A smaller geometric standard deviation for stratospheric coarse mode in MAM5-PSA means that there are fewer super-coarse aerosols within the population. " Although counterintuitive (adding a larger mode leads to smaller size distribution), this mechanism does seem like a possibility. But, there is no evidence shown to support this claim. Also, later (l393) it is pointed out that in MAM5, certain growth processes are not operational in the stratospheric course mode. At line 399, both processes are referenced as reasons for the sulfate having a longer lifespan in MAM5-PSA than MAM4. Again, both reasons seem possible, but is their evidence to support the idea that they are important?
**Response:**
Thank you for the comment. As of right now, no observational data of the size distribution of Pinatubo's sulfate aerosols exists on a global scale, with only local measurements as seen in Fig. 6. A reduction in the geometric standard deviation of dust aerosols leading to a longer lifespan in CESM and E3SM is also reported by Wu et al. (2020).

Similarly, observational data is also lacking for process understanding of stratospheric sulfates. Our goal for this study is to use our existing knowledge of aerosol processes to deduce what may occur within the stratosphere.

We have added a sentence to point this out below Fig. 1:
"The reduction in the geometric standard deviation (from 1.6 in the accumulation mode to 1.2 in the stratospheric coarse mode) is a method previously used in Wu et al. (2020) to lengthen the lifespan of aerosols."

12. L376: Yes, Baran and Foote have subtracted a baseline value away from the sulfate mass loadings they compute for Pinatubo. This does not mean their values are stratospheric burdens, and they do not use that language. This is because the tropospheric burden after the Pinatubo eruption can and likely is elevated compared to its background state.
**Response:**
Thank you for pointing out that the calculated sulfate burden by Baran and Foot (1994) also includes the troposphere. We have clarified this in the first paragraph of section 3.2:
"The derived sulfate burden from HIRS is calculated by taking the differences in the signal between the post-eruption period and an aerosol-free background, which means that it also includes tropospheric sulfate; however, tropospheric sulfate from the Pinatubo eruption has a lifespan of only several days and contributes only negligibly to the total burden past the first few days (Ukhov et al., 2023)."

13. Fig 10: Please update panel titles in figure to remove word "normalized"
**Response:**
Thank you for the reminder; we have revised the panel titles as suggested.

[Figure]

14. Fig 11 and related text: it is important to be clear that AVHRR is a total AOD anomaly, while GloSSAC reports a stratospheric AOD. Similar to the HIRS data, AVHRR's AOD likely includes some tropospheric anomaly component in the first months after the eruption.
**Response:**
We have clarified this within the discussion of Fig. 11.
"It is also worth noting that GloSSAC data includes only stratospheric AOD, and is missing tropospheric AOD anomalies that are included in AVHRR and the model results. Similar to the sulfate burden as discussed in section 3.2, the tropospheric AOD from the Pinatubo eruption is not significant compared to the stratospheric values."

15. L690: What results are you citing here, observations, models, etc? If models, what kind of models are they? The range of model results in that study appears to be 8-23 months, but that is for the decay of sulfate burden, not AOD. The two can be different as the size distribution changes with time.
**Response:**
We are citing a multi-model study by Quaglia et al. (2023). To avoid confusion and mismatching comparisons we have removed this sentence.

16. L699ff: please be clear through this section whether you are comparing SW fluxes or SW+LW total fluxes. The discussion starts with reference to solar (i.e., SW) fluxes, but the comparison to results from Hansen et al. (2005) refers to their total (SW+LW) flux anomaly values, with a peak of about -3.0 W/m^2.
**Response:**
Thank you for pointing this out. We have separated the discussion of solar fluxes and total (SW+LW) fluxes into two separate paragraphs to avoid confusion. The first paragraph contains only discussion of solar fluxes, while the second paragraph discusses the total flux.

We also made it clearer in the text by adding "(shortwave plus longwave)" flux anomaly at the beginning of the second paragraph.

**References:**

Baran, A. and Foot, J.: New application of the operational sounder HIRS in determining a climatology of sulphuric acid aerosol from the Pinatubo eruption, Journal of Geophysical Research: Atmospheres, 99, 25673-25679, 1994.

Emmons, L. K., Walters, S., Hess, P. G., Lamarque, J.-F., Pfister, G. G., Fillmore, D., Granier, C., Guenther, A., Kinnison, D., and Laepple, T.: Description and evaluation of the Model for Ozone and Related chemical Tracers, version 4 (MOZART-4), Geoscientific Model Development, 3, 43-67, 2010.

Guo, S., Bluth, G. J., Rose, W. I., Watson, I. M., and Prata, A.: Re-evaluation of SO2 release of the 15 June 1991 Pinatubo eruption using ultraviolet and infrared satellite sensors, Geochemistry, Geophysics, Geosystems, 5, 2004.

Mills, M. J., Schmidt, A., Easter, R., Solomon, S., Kinnison, D. E., Ghan, S. J., Neely III, R. R., Marsh, D. R., Conley, A., and Bardeen, C. G.: Global volcanic aerosol properties derived from emissions, 1990–2014, using CESM1 (WACCM), Journal of Geophysical Research: Atmospheres, 121, 2332-2348, 2016.

Quaglia, I., Timmreck, C., Niemeier, U., Visioni, D., Pitari, G., Brodowsky, C., Brühl, C., Dhomse, S. S., Franke, H., and Laakso, A.: Interactive stratospheric aerosol models' response to different amounts and altitudes of SO 2 injection during the 1991 Pinatubo eruption, Atmospheric Chemistry and Physics, 23, 921-948, 2023.

Ukhov, A., Stenchikov, G., Osipov, S., Krotkov, N., Gorkavyi, N., Li, C., Dubovik, O., and Lopatin, A.: Inverse modeling of the initial stage of the 1991 Pinatubo volcanic cloud accounting for radiative feedback of volcanic ash, Journal of Geophysical Research: Atmospheres, 128, e2022JD038446, 2023.

Wu, M., Liu, X., Yu, H., Wang, H., Shi, Y., Yang, K., Darmenov, A., Wu, C., Wang, Z., and Luo, T.: Understanding processes that control dust spatial distributions with global climate models and satellite observations, Atmospheric Chemistry and Physics, 20, 13835-13855, 2020.

---

## Author Response (AR4)

Response to the handling editor:

Thank you for pointing out the discrepancy. We have changed the wording in that section:

"…however, the tropospheric sulfate burden from the Pinatubo eruption is relatively small compared to the stratospheric burden. Results from Ukhov et al. (2023) suggest that the eruption contributed about 0.03 to tropospheric AOD two weeks after the eruption, compared to about 0.13 to the stratospheric AOD, with tropospheric sulfate having a lifespan of roughly one month."

---

## Author Response (AR5)

**Response to Comments of Reviewer #1**

**Title:** Size-resolved process understanding of stratospheric sulfate aerosol following the Pinatubo eruption

**General comments**:

The paper is mostly well written with a specific purpose but there are difficulties. The first difficulty is the limited comparisons with observations, just two sets of observations and two figures. Both figures raise questions about the comparison, but little discussion is given to the disagreements. A lot of papers have been published describing the post Pinatubo aerosol from a variety of instruments. Why aren't additional comparisons made with a much wider set of data? The model has pretty fine resolution, so it should not be limited to comparisons to measurements with global coverage. The authors spend a lot of energy comparing the various size distribution modes from the different model configurations, but make no attempt to compare any of these size distributions to observed size distributions. The authors offer no explanation for their limited comparison with observations.

Thanks to the referee for the helpful comments and constructive suggestions. We have revised the manuscript carefully and the point-to-point responses are listed below.

With regards to comparisons with observational data, we have already compared our results against stratospheric sulfate burden from HIRS observations (Figure 4 of revised manuscript) and AOD against AVHRR observations (Figure 8 of revised manuscript). Other sources of observational data are available, such as TOA radiative flux with ERBS and aerosol size comparisons with WOPC and SAGE. However, Brown et al. (2024), another study regarding the simulation of Pinatubo in E3SMv2 (the same model as we used in this work), has mostly already covered these comparisons. Their PA experiment is extremely similar to the MAM4SC experiment in this study, while their SPA experiment is extremely similar to MAM5SC excluding the addition of an independent stratospheric coarse mode. Neither the addition of the new mode or the use of a more complex chemistry scheme is meant to significantly alter model output of TOA radiative flux, and so we do not think it is necessary to repeat such comparisons in this work, and instead refer the reader to Brown et al. (2024).

Since a new stratospheric coarse mode is added in this work, we have included a new Figure 6 (see below) in the revised manuscript to compare simulated volume-size distribution against the observations from WOPC following your suggestions. It can be seen that MAM5FC did better capture the coarse mode volume (or mass) of sulfate aerosol in 1992 and 1993.

Furthermore, we have added a global mean AOD anomaly comparison between model simulations and two satellite-derived AOD datasets, AVHRR and GloSSAC (Figure 9 in the revised manuscript, see below). These two observations provided global coverage during the Mt. Pinatubo eruption. Due to limitations in the onboard instruments, AVHRR is more sensitive to the rapid AOD increase caused by volcanic eruptions but becomes less accurate when AOD values fall below 0.01 (Russell et al., 1996; Quaglia et al., 2023). Conversely, GloSSAC measurements become saturated when AOD exceeds 0.15, but are accurate when AOD values are relatively small (Thomason et al., 2018). This justifies using AVHRR and GloSSAC to quantify the upper and lower bounds of the AOD changes caused by Mt. Pinatubo. Consequently, Figure 9 in the revised manuscript has been added to evaluate the performance of different aerosol-chemistry schemes. We assessed the time evolution and e-folding time of the simulated global AOD anomaly against satellite observations (AVHRR and GloSSAC). The MAM5FC and MAM5SC simulations generally produced

reasonable AOD peaks and decay patterns, while MAM4FC and MAM4SC tended to underestimate AOD strength and overestimate the decay rate.

[Figure]

**Figure 6: Comparisons of stratospheric aerosol size distributions from MAM5SC and MAM5FC with observations from WOPC for 1991–1993. WOPC launches are samples taken from the 18 km measurements and matched to the nearest model height and grid cell over Laramie, Wyoming (41.3° N, 105° W).**

[Figure]

**Figure 9: Comparison of AOD anomaly between simulations and observations. The left panel shows the time evolution of monthly AOD anomaly values from simulations and AVHRR and GloSSAC observations, while the right panel shows the time evolution of normalized AOD anomaly values.**

**Major concerns/questions:**

1. 119-120 In terms of size order aren't the modes: Aitken, accumulation, coarse, rather than accumulation first? If so then they should be listed in that order.
   **Response:**
   Thank you for pointing this out. We have revised the text to be ordered by size.

2. Fig. 1 What happens to particles in MAM4 when they exceed Dg_hi (0.48 µm), which is not that large for sulfate particles following Pinatubo? See e.g. Deshler et al., GRL, 1992.
   **Response:**
   In MAM4 renaming is not turned on, therefore the geometric mean diameter is not allowed to exceed Dg_hi. The model will instead increase the aerosol number to maintain conservation of mass while also maintaining a maximum geometric mean diameter.

3. 178-179 The parenthetical clause is so long the reader has lost the thread as to what limits the aerosol formation rates.
   **Response:**
   We have restructured this sentence to make it more readable.
   "Most importantly, OH radicals (as well as other oxidants) are prognostically calculated rather than using prescribed values. This allows the model to represent the localized depletion of OH radicals due to the injection of large amount of $SO_2$, bottlenecking aerosol formation rates."

4. Fig. 2 and its discussion. What is the explanation for the aerosol in the Southern Hemisphere, which appears at most longitudes almost simultaneous with the Pinatubo eruption, particularly in MAM5FC? The presence of this aerosol clearly above background should be acknowledged and if possible explained.
   **Response:**
   The issue with aerosol in the Southern Hemisphere is caused by the color bar, because the background values for sulfate aerosol are just slightly larger than the upper limit of the white color. To avoid confusion, we have replotted it with a different color bar (see below).

[Figure]

**Figure 2: Simulated distributions of sulfate aerosol concentrations (kg/kg) at 53 hPa for days 3, 9 and 15 after the eruption of Pinatubo, respectively, for experiments MAM4SC (first row), MAM4FC (second row), MAM5SC (third row), and MAM5FC (bottom row).**

5. 284-286 Why does the smaller geometric standard deviation in MAM5 lead to more persistence? Is it because the particles are smaller in MAM5 compared to MAM4 and therefore less sedimentation? In any case there should be a sentence to describe the physics involved.

**Response:**

The removal rate of aerosols from the stratosphere in E3SM depends heavily on aerosol size, with larger aerosols having a significantly shorter lifespan. This is because the Stokes' settling velocity is roughly proportional to the square of aerosol diameter. With all other factors being identical, when the geometric standard deviation is smaller (i.e. smaller particles are less small, and larger particles are less large), a smaller portion of the aerosol population is large enough to be removed more quickly. We have added a brief explanation and a reference to the Seinfeld and Pandis textbook (Seinfeld and Pandis, 2016).

6. Fig. 4 and its discussion. The discussion mostly consists of describing the figure providing specific dates and sulfate burden peaks for the various modes. While perhaps these details are interesting they are all available in the figure for the interested reader. More interesting for the reader would be more discussion of the model / measurement discrepancies. Why do all models except MAM4FC over estimate the peak observed sulfur burden by 20%? Are the HIRS data reliable at the peak or are they suffering a saturation problem? Why does HIRS fall off so much faster than the MAM5 models in 1993? Additional interesting detail would be the range of median sizes involved in the large particle mode.

**Response:**
There are several possible reasons for the disagreement. The first is that HIRS results from Baran and Foot (1994) are not particularly accurate outside of the tropical areas due to errors introduced from the background signal (this is acknowledged in their results and discussion section), and as time passes sulfate aerosol is transported towards the poles.

In general, MAM5FC is able to accurately reflect the observed burden prior to 1993. With respect to the slower fall off in MAM5 models relative to HIRS in 1993, further improvement can be done in our future study. As can be seen in the newly added Figure 6 above, the stratospheric coarse mode burden in MAM5FC consists of particles somewhat smaller than observations, leading to a longer lifespan.

7. Because MAM4 doesn't have a coarse mode and MAM5 has a very narrow accumulation mode, both Figs 4 and 5 show the same thing, no contribution in the accumulation mode (or very little) from MAM5 and no contribution in the coarse mode from MAM4. Why then show the accumulation mode at all? Combine the MAM4 accumulation mode and MAM5 coarse mode into one figure. MAM5 accumulation mode could be included as a dotted line in the coarse mode plot. Then the two models can be more easily compared in terms of sulfate burden carried in the large particle mode.

**Response:**
The stratospheric coarse mode is unique to MAM5 which does not exist in MAM4, so we would prefer to keep the plots separate in order to emphasize this. However, we have changed the line styles in the figures according to your suggestions here and in your question #16 (we consistently use blue color for MAM4, red for MAM5, solid line for FC, and dashed line for SC). See revised Figure 5 below as an example.

[Figure]

**Figure 5: Simulated stratospheric sulfate burden (in Tg S) divided by mode and latitudinal region. The vertical lines represent the Pinatubo, Cerro Hudson, Spurr and Lascar eruptions respectively.**

8. Fig. 5 Same comments as Fig. 4 but in addition the ordinates should all be the same scale (0-4 Tg S), so the relative contributions from the different latitude zones can be seen directly. Without that the reader immediately wonders about the Southern Hemisphere signal which seems to persist at high levels. The label on the ordinate is wrong. It should be Sulfur burden (Tg S). Label the rows in some other way or describe them in the figure caption. Again combine accumulation and coarse mode into one plot then there will only be three rows. The latitudes of the eruptions should be listed in the figure caption.
   **Response:**
   Thank you for the suggestion. While using the same scale does have some merits for comparison of magnitude, our original intention is to focus on the aerosol microphysical processes (for example, how long they last in each different mode, etc.). If we set the scale to be identical across the figures, some features would not be very readable, such as Aitken mode with very small burdens.

9. 331-332 What is the reason for this oscillation, when the sulfur burden is declining everywhere else? A somewhat similar oscillation, but offset, is observed in the Northern Hemisphere. Why does the sulfur burden persist to 1994 in the Southern Hemisphere while it decays everywhere else?
   **Response:**

As mentioned in our responses to your question #6, further improvement can be done in our future study. As can be seen in the newly added Figure 6 above, the stratospheric coarse mode burden in MAM5FC consists of particles somewhat smaller than observations, leading to a longer lifespan. If we adjust the size parameters to have larger sulfate particles, we would see larger sedimention along the path of transport and hence smaller burden and oscillation in high latitudes.

10. 341-343 This point would be a lot clearer for the reader if the ordinate scale on all plots was the same. But this statement, "Starting from 1993, the stratospheric burden south of 30 S in MAM5 begins to make up at least half of the total between 80 S and 80 N (about 0.8 Tg S out of the total of 1.7 Tg S)", doesn't make sense. Between 80 S and 80 N includes 30 S – 30 N, so the total sulfur burden in early 1993 for MAM5FC is ~4.1 Tg S.
    **Response:**
    We prefer to keep the current scales for the reasons described in our responses to your question #8.
    Thank you for pointing out the error in numbers. We have corrected it as "(0.8 Tg S out of the total of 4.1 Tg S)".

11. 349-350 Suggest rephrasing to. "The tendencies are the integrals over three-dimensions of: all longitudes, …"
    **Response:**
    Text has been rephrased as suggested:
    "The tendencies are integrated over the three dimensions of: all longitudes, the above-mentioned latitude range, and vertically above the tropopause."

12. Fig. 6 is problematic. There are too many rows making the figure so small that most readers have to blow it up to see it. This could be fixed by breaking it into two figures: the first containing the first 4 rows, the next containing the last 3. Another suggestion is to combine the right and left panels into a single plot with separate ordinates on the right and left. Since the left shows a rate and right shows an accumulation the lines will generally not overlap but rather complement each other. Again the ordinate labels are incorrect. They should be tendency (kg/s) for the left plots and their integrals (or cumulative) (Tg S) for the right. Include the name of the row as a label in each plot. The labels RNMxx are too tied to the inner workings of the model, "renaming". But physically what is happening? The particles are growing to the next largest size distribution mode. If labels were added to the plots to identify them RNMaa could become Aitken->Accumulation mode and RNMasc Accumulation->Coarse mode.
    **Response:**
    We actually present the units at the top of the figure for both columns, since each column shares the same unit.
    Renaming is not the only way for aerosols to transit between modes in E3SM. Please see definitions of all processes in our response to your question #13 below.

13. Fig 6 e, f) shows the growth from Aitken to accumulation mode presumably by several processes including growth by condensation and coagulation, correct? If that is the case then why are the ordinate scales on COAG so much larger than for RNMaa? Both are showing

mass loss rate and total mass lost. It seems it should be the other way around with COAG less than RNMaa. Why is this one process coagulation singled out for a special plot?

**Response:**

In the model, condensation, coagulation and renaming are each different aerosol processes. Condensation/evaporation represents the gas-aerosol mass exchange, e.g. transition from $H_2SO_4$ gas to sulfate aerosol. Coagulation is the process of multiple smaller particles colliding into each other to form larger particles. Renaming is not a "real" physical process and is instead an internal calculation within the model where particles that have grown sufficiently large are moved from one aerosol mode to another. Coagulation and renaming are two separate ways for aerosols to transit from the Aitken mode to accumulation mode; coagulation is not a subset of renaming or vice versa.

Because COAG and RNM are separate mechanisms, we think it is better to keep them separate in figures and discussions. A description of what renaming and condensation represent are in the model overview in Part 2.

14. Fig 6 i – n) Condensation? Why are these processes now called condensation? Condensing from what? Weren't these earlier called renaming, which is also not that helpful or descriptive. Isn't this particle growth from one mode to another? Ordinate problems again. What is on the left and right ordinates? Is it again rate (kg/s) and cumulative mass (Tg S)? The reader doesn't know and the figure caption does not help.

    **Response:**

    See our responses to your question #13. We have now clarified the terminology in figure captions.

[Figure]

**Figure 7: The relevant tendencies for each mode in each experiment in the stratosphere between 80°S and 80°N. See Figure 1(a) for description of processes. Positive values represent gained mass from the associated microphysical process from the perspective of the aerosol mode in question. NUCL represents the aerosol mass gain for Aitken mode due to nucleation (i.e.,**

aerosol formation), COAG represents Aitken mode mass loss due to coagulation into the accumulation mode, RNMaa represents Aitken mode mass loss due to renaming into the accumulation mode, RNMasc represents the gain in mass for stratospheric coarse mode in MAM5 due to renaming from the accumulation mode. The left column plots represent the tendency values over time, while the right column represents the cumulative mass change due to the associated microphysical process (i.e. tendency integrated over time). The bottom three rows represent the condensation tendencies for each mode. The vertical lines represent the Pinatubo, Cerro Hudson, Spurr and Lascar eruptions respectively.

15. 351-415 Again the figure discussion consists primarily of describing the figure, pointing out maximum values and dates when they occur. Relatively little is describing what can be learned from the figure which is the importance of the figure. Here and elsewhere, if these dates and amounts are particularly important organize them into a table. Then they could really be compared. It is not clear how listing them in the text helps the reader.
    **Response:**
    Following the reviewer's suggestion, we have simplified this paragraph, trying to remove some of the numbers and to emphasize the most important information of the figure.

16. Figs. 4-8, 11 The plotting for these figures could be made much more intuitive, so the reader doesn't constantly have to refer to the legend to remember which is which. It would be quite easy to do. Use one color for MAM4 and one for MAM5, then one line style (e.g. solid) for FC and another (dashed or dotted) for SC. Then each figure can be immediately understood without referring to the legend but once.
    **Response:**
    Thank you for the suggestion. We have changed the line styles in these figures to consistently use blue color for MAM4, red for MAM5, solid line for FC, and dashed line for SC).

17. Figs 7 and 8 suffer from the same problem as Fig. 6. There are too many panels and they are too small. What are Figs. 7 and 8 adding to what we learned from Fig. 6? Are all these rows necessary? Which ones are the most informative?
    **Response:**
    Figure 7 (previously Figure 6) indeed has too many panels to show processes for all the modes. These are important information for process understanding. We could split the figure into two parts but we would like to keep it for now and discuss it later with the journal editorial team.

18. 420-430 In fact the discussion of Figs. 7 and 8 acknowledges that not much new is added. "The same patterns as above apply between 30 S and 30 N. Above 30 N and below 30 S the same signal from the Pinatubo is still present, though slightly delayed due to the time that it took for the aerosol to transport poleward. Signals from other eruptions (Spurr and Lascar) are also present" Then a few interesting differences are discussed. Just show the interesting panels and combine Figs 7 and 8 into one figure with the few interesting panels.
    **Response:**
    We had discussions regarding condensation across different latitudinal regions in the text. However, following your suggestions, we have moved the original Figures 7 and 8 to the supplementary.

19. 503 condensation? Same questions as above. What does this mean?
    **Response:**
    See our responses to your question #13.

20. 503-505 What is the reason for quoting these numbers? How will the reader use such information rather than the already stated comparison about the difference rates? Too detailed.
    **Response:**
    We have cut out the specific numbers and left the qualitative comparisons.

21. 506 Here COAG is separated from RNMaa, but aren't both processes doing the same thing? There is only the transition from Aitken to accumulation. Is it important how it happens? If so why isn't that mentioned earlier?
    **Response:**
    See our responses to your question #13.

22. 509 deposition? Does this mean sedimentation out of the stratosphere? Generally deposition refers to losing aerosol due to contact with a surface.
    **Response:**
    "deposition" has been changed to "sedimentation".

**Response to Comments of Reviewer #2**

**Title:** Size-resolved process understanding of stratospheric sulfate aerosol following the Pinatubo eruption

**General comments**:

I think the article is well written and a nice advancement to stratospheric aerosol modeling but may be better suited in its current form for a journal such as Geoscientific Model Development. While the differences in aerosol loading due to model configuration are made clear, it is left to the reader to interpret how this improves understanding of atmospheric chemistry or physics. The authors provide in-depth discussion on the relative importance of coagulation, nucleation and renaming/growth in the model, but have little discussion on the physical processes this may help resolve. Similarly, few comparisons are made with measurements for AOD, particle size or radiative flux, with no discussion given to possible sources of disagreement, or what implications these results may have for observations (e.g. the assumptions going into the HIRS results used here). Personally, I think addressing any of these points would help expand the applicability of the paper to a more general audience.

Thanks to the referee for the helpful comments and constructive suggestions. We have revised the manuscript carefully and the point-to-point responses are listed below.

As shown by the title of the manuscript, this work aims to provide in-depth discussion on the relative importance of coagulation, nucleation and renaming/growth in stratospheric sulfate formation.

With regards to comparisons with observational data, we have already compared our results against stratospheric sulfate burden from HIRS observations (Figure 4 of revised manuscript) and AOD against AVHRR observations (Figure 8 of revised manuscript). Other sources of observational data are available, such as TOA radiative flux with ERBS and aerosol size comparisons with WOPC and SAGE. However, Brown et al. (2024), another study regarding the simulation of Pinatubo in E3SMv2 (the same model as we used in this work), has mostly already covered these comparisons. Their PA experiment is extremely similar to the MAM4SC experiment in this study, while their SPA experiment is extremely similar to MAM5SC excluding the addition of an independent stratospheric coarse mode. Neither the addition of the new mode or the use of a more complex chemistry scheme is meant to significantly alter model output of TOA radiative flux, and so we do not think it is necessary to repeat such comparisons in this work, and instead refer the reader to Brown et al. (2024).

Since a new stratospheric coarse mode is added in this work, we have included a new Figure 6 to compare simulated volume-size distribution against the observations from WOPC following your suggestions. It can be seen that MAM5FC did better capture the coarse mode volume (or mass) of sulfate aerosol in 1992 and 1993.

We have also added a global mean AOD anomaly comparison between model simulations and satellite-derived AOD datasets, AVHRR and GloSSAC (Figure 9 in the revised manuscript), which provided global coverage during the Mt. Pinatubo eruption. AVHRR is more sensitive to rapid AOD increases caused by eruptions but becomes less accurate for AOD values below 0.01, while GloSSAC is accurate at lower AOD values but becomes saturated above 0.15. These observations help quantify the bounds of AOD changes from Mt. Pinatubo. The MAM5FC and MAM5SC simulations showed reasonable AOD peaks and decay patterns, while MAM4FC and MAM4SC

tended to underestimate AOD strength and overestimate the decay rate.

**Major concerns/questions:**
1.  Line 50-65: This geoengineering section seems a bit out of place to me. I'm sure this work has implications for geoengineering studies, but no indication of exactly what those may be is provided. I recommend clarifying the link to this work or removing this paragraph. Perhaps the geoengineering discussion and the link to this work would be better placed in the discussion/conclusion?
    **Response:**
    Geo-engineering is commonly cited as a motivation into the simulation of volcanic aerosols/stratospheric sulfates, e.g. Mills et al. (2017) introduction and Tilmes et al. (2022).

2.  Line 415: "MAM4 also generally has stronger nucleation and coagulation processes than MAM5." From Figure 6 it isn't clear whether MAM4FC has greater coagulation tendencies for physical reasons or if it is just due to the increased NUCL. This is discussed in the conclusion of the paper (Line 503), but I think should be mentioned here. I would suggest rewording to something like: "MAM4 also generally has stronger nucleation than MAM5, and due to these higher concentrations, increased coagulation processes as well" Or, if there are other reasons for increased COAG then this should also be discussed.
    **Response:**
    Thank you for the suggestion. A short explanation has been added to the sentence: "MAM4 generally has stronger nucleation leading to higher Aitken mode concentrations, and therefore also has stronger coagulation processes than MAM5."

3.  Line 490: "large differences in both the temporal variations and the spatial distributions of sulfate concentrations" It is difficult to tell from Figures 7/8, but in Figures 9/10 there doesn't appear to be much change in spatial distribution. Both MAM4 and MAM5 show large increase in the tropics and later transport to the NH. Some expansion on the spatial differences the authors are referring to would be welcome.
    **Response:**
    This is primarily referring to Figure 3, where SC experiments had sulfate concentrations at a noticeably higher altitude than FC. This is mentioned further on in the paragraph: "Vertically, sulfate distributions were generally at lower altitudes in FC compared to SC…"

4.  Line 521-522: If the use of full chemistry and MAM5 helped improve agreement with AVHRR, why is the TOA flux more comparable to Brown (2024) and Mills (2017) results than the MAM4 version? Is this related to the geometric standard deviation that was used?
    **Response:**
    The fundamental changes made to the aerosol module are similar across MAM5 in this work, the altered MAM4 scheme in Brown et al. (2024), and the altered scheme in Mills et al. (2017): they all allow sulfate to rename into coarse mode and adjusting some of the coarse mode parameters (e.g., reduced the geometric standard deviation of the coarse mode in their altered MAM4 to that as used in the stratospheric coarse mode in MAM5 of this work) to better fit observational data. On the contrary, MAM4 in this work does not allow sulfate to rename into coarse mode (i.e., continued increase in sulfate mass leads to an increase in the accumulation mode number concentration, rather than a transition from accumulation mode mass to coarse

mode mass). As a result, the TOA flux from MAM5 agrees closer to Brown (2024) and Mills (2017) than that from MAM4, and this is the expected result.

   In the second to last paragraph of the introduction, we have added the following sentences:
   "Brown et al. (2024) also simulated the Pinatubo eruption using E3SMv2, the same version of the earth system model as this study. Their prognostic aerosol simulations use the model's default "simple" chemistry, the same as our SC experiments. One of their experiments, E3SMv2-PA, used the default MAM4 (i.e., nearly identical to our MAM4SC experiment). Their other prognostic aerosol experiment, E3SMv2-SPA, used a revised MAM4, which treated dust, sea salt and stratospheric sulfate using the same coarse mode parameters, such as size range and geometric standard deviation, specifically chosen to best represent stratospheric sulfate properties, resulting in erroneous coarse mode dust and sea salt aerosol concentrations. To avoid these problems, this study establishes a fifth aerosol mode (MAM5-PSA), the stratospheric coarse mode, to specifically handle stratospheric sulfate. The coarse mode remains the same as the original MAM4 to handle dust and sea salt."

   In the second paragraph of the methodology section, we have added the following sentences:
   "In E3SMv2 MAM4, the coarse mode was primarily intended for coarse sea salt and dust, as well as other aerosol species resuspended from raindrop evaporation, with a geometric standard deviation of 1.8, and stratospheric sulfate aerosol cannot enter the coarse mode through renaming. Renaming is a process in which aerosol particles that grow larger than a given threshold via condensation and coagulation are transferred from one mode to another. To avoid these problems, this study establishes a fifth aerosol mode (MAM5-PSA), the stratospheric coarse mode, to specifically handle stratospheric sulfate. The coarse mode

remains the same as the original MAM4 to handle dust and sea salt. The newly added stratospheric coarse mode follows the same size parameters as the revised coarse mode in Mills et al. (2016) and Brown et al. (2024) to better reflect stratospheric sulfate lifespans, but handles stratospheric sulfate separately from coarse mode sea salt and dust. A portion of sufficiently large accumulation mode sulfate aerosols are permitted to transfer to the stratospheric coarse mode through renaming. It should be noted that the upper limit for the accumulation mode size range is larger than the lower limit for the stratospheric coarse mode size range, leading to potential size overlaps between the two modes' aerosol populations."

3. Please note that the HIRS sulfate burden observations are not of "stratospheric aerosol" as stated in your reply to Referee 1, the HIRS product is a total column burden. Even a burden anomaly can contain anomalous sulfate mass in both the stratosphere and troposphere.
**Response:**
The observed sulfate burden that we use in this study is derived by Baran and Foot (1994), rather than the direct product of HIRS. As described in that paper, a background signal (including that from tropospheric sulfate) has already been subtracted in order to calculate the stratospheric burden.

4. The statement "Neither the addition of the new mode or the use of a more complex chemistry scheme is meant to significantly alter model output of TOA radiative flux, and so we do not think it is necessary to repeat such comparisons in this work" confused me, as you do show TOA flux anomalies, and these results seem to show that changes in the mode structure and chemistry may indeed affect the TOA radiative fluxes. Please clarify in your revised responses to the referee comments.
**Response:**
Thank you for pointing this out. As we have described in the response to Comment #2, MAM4SC is nearly identical to the E3SMv2-PA experiment in Brown et al. (2024). Since they have already compared the net radiative flux of their E3SMv2-PA experiment against observations, there is no need to repeat this for our MAM4SC experiment.

As can be seen from the net TOA flux comparison in Brown et al. (2024), from July 1991 to the end of 1992, observed net radiative flux anomalies are within the range of -3.8 to -1 $W/m^2$. Their E3SMv2-PA's simulated values are in the range of -2 to +1.5 $W/m^2$, while for our MAM5FC experiment, they are in the range of -2.8 to -1.3 $W/m^2$. In terms of net radiative flux anomaly, MAM5FC agrees more closely with the ERBS observations. We have clarified this in the response to referee #2.

[Figure]

**Figure R1: screenshot of Figure 4 in Brown et al. (2024), a comparison of their E3SMv2-PA simulated net radiative flux anomalies (solid blue line) and ERBS observations (solid black line).**

[Figure]

**Figure 13: Global mean net radiative flux anomaly in the four experiments, where negative values represent a net upward flux. The anomaly is the difference with the corresponding model experiment with the volcanic emissions shut off.**

5. The addition of the comparison to the OPC data is welcome, but it seems to me that you are missing a chance to justify the addition of the course mode to the model by not including the MAM4 accumulation mode on these plots. It is also not clear to me how the results shown support the statement that "MAM5FC did better capture the coarse mode volume (or mass) of sulfate aerosol in 1992 and 1993."
   **Response:**

[Figure]

**Figure 4: (a) Comparisons of simulated stratospheric sulfate burdens (Tg S) between 80° S and 80° N from the four experiments with HIRS observations for years of 1991-1995. Black line depicts the estimates made from HIRS data in Baran and Foot (1994), with the black bars representing the margin of error. (b), (c), and (d) are similar comparisons for the Aitken mode, accumulation mode and stratospheric coarse mode respectively (renaming from accumulation mode into the coarse mode does not exist in MAM4).**

We did not include MAM4 in Figure 6 of the manuscript because we already know from Figure 4(a) (included above) that its sulfate lifespan is too short, and is not expected to be comparable to observations. The red curves (corresponding to MAM5FC) in Figure 6 cover an area closer in size to the black curves (representing the observations), compared to the blue curves (MAM5SC), in the four plots from 1992 and 1993. The integrated area below the curve in Figure 6 represents the total mass of that aerosol mode. However, both MAM5-PSA experiments have a smaller mean diameter than the observations, i.e. they are to the left of the black curve. We have added some explanation to aid understanding.

"In 1992 and 1993, the coarse mode mass (and volume) for MAM5FC is closer to WOPC observations than MAM5SC."

For reference we have added the mass distribution plots for MAM4 here (the observed coarse mode from WOPC). Note that the accumulation mode sulfates are actually larger than the stratospheric coarse mode sulfates in MAM5-PSA. As explained below in the response to comment 7, the largest accumulation mode aerosols are slightly larger than the smallest stratospheric coarse mode aerosols. It is also worth noting that aerosol size does not entirely reflect aerosol properties, as certain microphysical processes in E3SM are restricted by aerosol mode rather than size distribution. We've also added this figure to the supplementary.

[Figure]

**Figure S5: Similar to Figure 6 within the manuscript, but for the MAM4 experiments instead.**

6. Concerning Referee 1's comment "Fig. 1 What happens to particles in MAM4 when they exceed Dg_hi (0.48 μm), which is not that large for sulfate particles following Pinatubo? See e.g. Deshler et al., GRL, 1992." It is important to point out here in the manuscript whether or not the mode structure of MAM4 is inherently ill-equipped to represent stratospheric aerosol from a large eruption like Pinatubo. One needs to be careful because capping Dg_hi to 0.48 μm doesn't mean that is the largest size (since it's a distribution), but some words in the manuscript coming the parameters of the mode set up with what is known about Pinatubo aerosol is needed here to address the referee's comment.

**Response:**

Yes, Dg_hi is the maximum for the geometric mean diameter and not the sulfate particle size, because the aerosol population's size follows a lognormal distribution. We have clarified this in Figure 1's description in the methodology section: "The geometric dry mean diameter (Dg) is prognostically calculated, and can become larger or smaller than the initial prescribed Dg value in Figure 1(a) depending on microphysical processes. Dg_hi and Dg_lo are the upper and lower limits for the geometric dry mean diameter."

We have pointed out in the introduction that: "… MAM4, which treated dust, sea salt and stratospheric sulfate using the same coarse mode parameters, such as size range and geometric standard deviation, specifically chosen to best represent stratospheric sulfate properties, resulting in erroneous coarse mode dust and sea salt aerosol concentrations. To avoid these

problems, this study establishes a fifth aerosol mode (MAM5-PSA), the stratospheric coarse mode, to specifically handle stratospheric sulfate. The coarse mode remains the same as the original MAM4 to handle dust and sea salt."

7. Concerning the relationship between sedimentation rates and the mode structure of MAM4 and MAM5, the explanations given seem to oversimplify the issue, since it is not just the standard deviation of the mode that controls how many "very large" particles there are, but also the mode's mean radius. Therefore, it is not readily apparent that an accumulation mode with large SD will always have more large particles than a course mode with smaller SD since the course mode can have a larger mean radius. Evidence is required here to support the arguments made. Also, the presentation should be careful to remember that most readers will automatically assume that a course mode is larger in size than an accumulation mode. Here it seems that the modes overlap strongly and that is an important things to point out and discuss, in contrast to many modal aerosol schemes where the modes are well separated.

**Response:**
Thank you for pointing this out. In order to avoid confusion, we have rewritten the discussion of Figure 4 to clarify the roles of both the geometric mean diameter and the standard deviation:

"Starting in January 1992, the stratospheric burdens in MAM4 drop off more quickly compared to MAM5-PSA and MAM5FC agrees closely with HIRS until the end of 1992. In MAM4, sulfate in the accumulation mode grows until it reaches its maximum allowed geometric dry mean diameter of 0.48 microns (the Dg_hi value for the accumulation mode). Under such situations, the sulfate aerosol population increases in number (and therefore mass), but the size cannot increase any further. In MAM5-PSA, renaming is permitted, and the transfer from the accumulation mode to the stratospheric coarse mode begins when the geometric dry mean diameter is 0.40 microns (i.e. the Dg_lo value for the stratospheric coarse mode). Upon transfer to the stratospheric coarse mode, certain aerosol growth processes such as coagulation cease (as it is only allowed for the Aitken and accumulation mode in MAM), so the stratospheric coarse mode sulfates in MAM5-PSA tend to grow no larger in size, and the geometric dry mean diameter remains around 0.40 microns. In addition, the reduced geometric standard deviation of MAM5-PSA's stratospheric coarse mode (1.2) relative to MAM4's accumulation mode (1.6) also corresponds to a smaller proportion of super-coarse aerosols that are removed more quickly from the stratosphere, due to the settling velocity being roughly proportional to the second power of the aerosol diameter (Seinfeld and Pandis, 2016). These two reasons are responsible for the sulfate having a longer lifespan in MAM5-PSA than MAM4."

The size overlap has been explained in the methodology:

"A portion of sufficiently large accumulation mode sulfate aerosols are permitted to transfer to the stratospheric coarse mode through renaming. It should be noted that the upper limit for the accumulation mode size range is larger than the lower limit for the stratospheric coarse mode size range, leading to potential size overlaps between the two modes' aerosol populations."

8. In your replies you mention uncertainties in the HIRS burden outside of the tropics, but such

discussion would be useful to include in the manuscript.

**Response:**

Following your suggestion, we have added the following sentences in the beginning of Section 3.2, where we describe the results in Figure 4:

"Figure 4(a) shows the time series of the simulated stratospheric burden of sulfate between 80° S and 80° N, as well as the HIRS monthly observational data for stratospheric sulfate burden for the same latitude bands taken from Baran and Foot (1994). Note that the stratospheric burden derived from HIRS data assumes that the mean stratospheric temperature is 210K, which is increasingly inaccurate with more distance from the tropics."

9. Please adjust plot axes labels according to the suggestions of referee #1.

**Response:**

The plot axes labels have already been changed following the suggestion of referee #1.

10. You have added some information about "renaming", but given its centrality to the results, I suggest even more description is needed, and some discussion to help differentiate it from the physical processes that make aerosols grow and move from one mode to another. Why is renaming needed if the model includes these processes that grow particles? I assume it is important to include when the modes overlap in terms of radii included with the different modes. What controls the rate of renaming? Does it only act to increase size, or does it act in both directions?

**Response:**

Renaming is not a physical process per se, but transfers aerosols from one mode to another as a result of microphysical processes such as condensation and coagulation. The rate of renaming is controlled by the microphysical processes. It acts to increase aerosol size (not in both directions). We have considered seven different microphysical processes regarding the sulfate aerosol in the manuscript. The following explanation has been added to the methodology section to help the reader understand the differences between them:

- NUCL, or nucleation, is responsible for the formation of aerosols starting from the Aitken mode (nucleation mode aerosols are not represented in MAM).
- COAG, or coagulation, represents two smaller particles colliding to form a larger particle. MAM considers the intramodal and intermodal coagulation of the Aitken, accumulation and primary carbon modes. Intramodal coagulation reduces the number of the mode but leaves the mass unchanged. Intermodal coagulation between these modes and the coarse mode is not considered in MAM.
- AitkenCond, or condensation in the Aitken mode, leading to aerosol mass increase in the Aitken mode population. No matter how much mass is gained from condensation, the aerosols cannot grow into the accumulation mode directly, it MUST undergo renaming.
- AccumCond and StratoCoarseCond, condensation for the accumulation mode and stratospheric coarse mode, work similarly.
- RNMaa, or renaming from the Aitken to accumulation mode. Periodically, the size of the Aitken mode aerosol population is checked. If a specified threshold is reached (due to mass increase from the condensation and coagulation processes), then aerosol

number and mass are transferred from the Aitken mode to the accumulation mode.

- RNMasc, renaming from the accumulation mode to stratospheric coarse mode. No other microphysical process is allowed to produce stratospheric coarse mode sulfates, so this is their only source.

11. Figure 7: please confirm, are the quantities here still in terms of mass sulfur?
**Response:**
Thank you for pointing this out, we have clarified this in the figure caption that it is the total sulfate ($SO_4$) aerosol mass.

12. Figure 7: please address referee comments about the size of panels. There is a lot of information on these plots, and what is worth showing should be related to the conclusions one draws from the results. There is some lack of clarity in what exactly are the important conclusions a reader is supposed to draw from these plots, aside from that there are differences between the models.
**Response:**
We have carefully examined Figure 7 and feel that all of the panels are necessary to understanding the process. For the sake of clarity, we have separated Figure 7 into three parts, one for nucleation and coagulation, one for condensation, and one for renaming. A fourth part depicting the sum of condensation across all modes has been added to the supplementary for reference (Figure S6).

[revised manuscript text omitted]

13. The word "deposition" is still found at the end of page 23 and should be changed as stated in the replies to referee #1.
    **Response:**
    Thank you for pointing this out. We have replaced "deposition" with "sedimentation".

14. Concerning the "geoengineering" focus of lines 50-65, while I agree that geoengineering is a motivation for studying volcanic eruptions, the issue here is that the references in this paragraph do not actually support the first sentence, which states that "Previous studies have shown that the impacts of volcanic aerosol injections on climate are dependent on the latitude, aerosol type, and season of the injection of volcanic aerosols." While the geoengineering studies cited are not irrelevant, there are other more relevant studies which specifically look

at the impact of latitude, aerosol type and season on aerosol from eruptions like Pinatubo. It is essential that the introduction of this work do a better job of summarizing the existing literature on modeling of Pinatubo and Pinatubo-like eruptions. The review paper of Marshall et al. (2022) and references therein should provide a base upon which to build a more comprehensive literature review.

**Response:**
Following your suggestion we have included some literature more directly relevant to the simulation of volcanic eruptions:

"Numerous previous studies have examined volcanic effects on climate and possible influencing factors such as the location and timing of the eruption and total emissions. Using statistical emulation of output of the UK Met Office Unified Model coupled with the UK Chemistry and aerosol scheme (UK-UMCA), Marshall et al. (2019) found that for large $SO_2$ emissions of 10 to 100 Tg, the e-folding decay time for the global sulfate aerosol burden is most dependent on the latitude of the eruption, while the integrated global mean stratospheric AOD and net radiative forcing is most dependent on the total mass of the $SO_2$ emissions. Zhuo et al. (2024) used the Whole Atmosphere Community Climate Model Version 6 (WACCM6) to simulate volcanic eruptions on a similar scale to Pinatubo from Central America and Iceland under different ENSO initial states, finding that initial atmospheric conditions control the meridional transport of sulfur and halogens in the first month after the eruptions as well as further modulating the latitudinal distribution of sulfate aerosols, halogens, volcanic forcing and impacts. Toohey et al. (2019) analyzed ice-core-derived volcanic stratospheric sulfur injections and Northern Hemisphere summer temperature reconstructions from tree rings, finding that in proportion to their estimated stratospheric sulfur injection, extratropical explosive eruptions since 750 CE have produced stronger hemispheric cooling than tropical eruptions; stratospheric aerosol simulations with the MAECHAM5-HAM model suggested that this was due to the enhanced radiative impact associated with the relative confinement of aerosol to a single hemisphere."

15. Concerning "large differences in both the temporal variations and the spatial distributions of sulfate concentrations", please edit the sentence to make it clearer you are referring to the vertical distribution of aerosol.
    **Response:**
    Thank you, we have changed this to "large differences in both the temporal variations and the vertical distributions of sulfate concentrations".

16. Revised manuscript: end of paragraph 1. Please note, Kremser et al. (2016) states stratospheric aerosol is removed "through sedimentation and in air traversing the extratropical tropopause", so this citation is misleading. Both are important, it is not only sedimentation that removes aerosol. In fact, other studies (e.g., Hamill et al., 1997) argue that cross tropopause air mixing is much more important than sedimentation for the removal of stratospheric aerosol.
    **Response:**
    We have edited the manuscript to correctly reflect the conclusions of Kremser et al. (2016).
    "Eventually, these stratospheric aerosols are removed by entering the troposphere through sedimentation processes and cross tropopause air mixing."

17. Paragraph 2 quote a range for SO2 for Pinatubo of 15-18 Tg SO2, then on page 10 we are told simulations are run with 10 Tg SO2 injection. This is a significant difference and requires justification. Also, the uncertainty in the SO2 injection is a very important issue for discussion of the results and their implications.
**Response:**
Thank you for pointing this out. Mills et al. (2016) also ran with a 10 Tg injection. We use the same value for parity, and also because CESM and E3SM are similar enough that the same reasoning is relevant. As mentioned in their section 2.2:

"Our model does not currently simulate volcanic ash or water vapor emissions nor consequent ice sequestration of $SO_2$ and fallout of ash and ice particles, which took place mainly during the first 4–5 days of the 1991 Pinatubo eruption (Guo et al., 2004). In the model, we therefore reduce the mass of $SO_2$ emitted to 10 Tg of $SO_2$, which corresponds to the mass detected by TIROS Operational Vertical Sounder and TOMS 7–9 days after the beginning of the eruption, when more than 99% of the ash and ice particles were removed (Guo et al., 2004). This is what we consider to be the "climatically relevant" portion of the 1991 Mount Pinatubo $SO_2$ emissions for our model."

We have also added a sentence explaining this in the methodology section.

18. A major novelty of this work appears to be the OH replenishment mechanism. It would be useful to have more discussion of this—have other modeling studies included this mechanism? How confident are we of the chemistry? If there is confidence that replenishment occurs, what do the results here tell us about the importance of this process?
**Response:**
The original chemical reaction can be found in Atkinson et al. (2004), Table 1, reactions 86 and 87, though for simplicity the two reactions are merged into one inside the model. This reaction is present in and adapted from WACCM's chemistry scheme, which implies that it should be included in Mills et al. (2016), though it is not explicitly stated in their paper.

Our results indicate that the inclusion of the OH replenishment mechanism is important for the simulation of stratospheric sulfate aerosols. The simulations with FC better capture stratospheric sulfate burden (Fig. 4(a)), AOD (Figs. 10 and 11) and net radiative flux (Figs. 12 and 13).

19. Figure 4: Prior studies (e.g., Clyne et al., 2021) have suggested that OH depletion slows the rate of aerosol growth, leading to smaller particle size and therefore a longer lasting aerosol burden. The results shown here do not seem consistent with this theory. Discussion is warranted.
**Response:**
This appears to be partially true. The full chemistry experiments in our study, where OH depletion is possible, do have larger Aitken mode burdens and smaller accumulation and coarse mode burdens (see Figure 4), compared to the simple chemistry experiments where OH depletion is not possible. However, the differences in the Aitken mode quickly become insignificant relative to the stratospheric coarse mode after January 1992, and the aerosol

lifespan is largely reflected in the stratospheric coarse mode.

Looking at Figs. 7-9, it can be seen that the sum of the tendencies of renaming from accumulation mode into stratospheric coarse mode and condensation in the stratospheric coarse mode (i.e. RNMasc + StratoCoarseCond) is largely the same between MAM5SC and MAM5FC, suggesting that the produced mass of stratospheric coarse mode sulfates is similar (i.e. roughly the same proportion of sulfates made it into the stratospheric coarse mode in terms of mass, so lower OH concentrations do not seem to prevent sulfates from reaching the stratospheric coarse mode eventually), and the difference in lifespan can probably be attributed to sedimentation rate differences, which in turn might be influenced by many factors. For example, even though the total mass of the produced stratospheric coarse mode sulfates is similar, they might be produced in different geographical locations, or at different altitudes, some of which might encourage sedimentation more than others.

In summary, the difference in aerosol lifespan between the MAM5FC and MAM5SC experiments occur long after oxidation of $SO_2$ by OH radicals has ceased in the stratosphere, outside of new volcanic eruptions after Pinatubo. The difference in aerosol lifespan is decided by the sedimentation rate of the stratospheric coarse mode, since regardless of whether OH is interactive, about the same proportion of sulfates eventually reach the stratospheric coarse mode. This difference in sedimentation rates is likely due to a large number of different factors interacting with one another, and although MAM5SC and MAM5FC are originally differentiated from one another by their handling of OH, the large span of time between the original formation of sulfate and the eventual sedimentation makes it difficult to directly attribute differences in the former to that of the latter.

20. Sect. 3.4, subtracting the climatology from 1989 produces an anomaly: this is not a normalization. Text here and labels on figure 8 must be corrected.
    **Response:**
    Thank you, we have corrected the terminology.

21. Page 21: here it is stated "The e-folding time, calculated as the time between the maximum and the 1/e value, was 13 months in AVHRR and 23 months in GloSSAC." Looking at Figure 9 b, these times appear to actually be the time between the eruption and the crossing of the 1/e line, not the time from the peak value to the crossing of 1/e. The difference is quite important, see e.g., Toohey et al., (2024).
    **Response:**
    The start of each line representing the normalized AOD anomaly already represents the peak value, except for GLOSSAC, which crossed 1/e about 16 months after the peak value; we have rectified this to "16 months in GloSSAC".

22. Page 22, you could here compare the rough magnitude of the global mean TOA radiative flux changes with those shown from observations in Fig 11 of Hansen et al. (2005).
    **Response:**
    Thank you for the suggestion. We have added a comparison.

    "It also agrees well with the simulation results from Hansen et al. (2005) up to early 1992,

which also reports a peak value of about -3.0 W/m$^2$ across their five runs, though the net radiative flux signal is slower to diminish around July 1992, concurring with the stratospheric sulfate burden depleting too slowly in MAM5FC."

23. The conclusion section includes a summary of results shown, and some discussion of possible caveats to the results, but lacks discussion of the implications of the results with reference to prior studies. This is an essential ingredient for papers in ACP (see https://www.atmospheric-chemistry-and-physics.net/policies/guidelines_for_authors.html) and should be improved.
**Response:**
We have heavily revised the conclusion section to emphasize the novelty of this study and the implications of our results with reference to prior studies. The updated paragraphs are listed below:

"In this work, we have implemented a fifth stratospheric coarse mode (MAM5-PSA) on top of the original four-mode version Modal Aerosol Module (MAM4) used in E3SMv2. We also consider a more complex "full chemistry (FC)" set-up, which includes the addition of a series of chemical reactions which allow the simulation of OH radical replenishment compared to "simple chemistry (SC)" with prescribed OH concentrations. We have carried out a series of simulations of the Pinatubo eruption for the time period of 1991-1993 to better understand the aerosol microphysical processes including nucleation, coagulation and condensation. We have also compared model simulation results with measurements of stratospheric sulfate burden, aerosol size, AOD and net radiative flux.

[revised manuscript text omitted]

24. Referee #2 points out that the subject matter of the paper is technical in nature. A reminder that research articles in ACP "must include substantial advances and general implications for the scientific understanding of atmospheric chemistry and physics." Given the results and conclusion of the paper focus on the representation of aerosol in models, and not general scientific understanding, I strongly suggest the manuscript type be changed to technical note. https://www.atmospheric-chemistry-and-physics.net/about/manuscript_types.html
**Response:**
We respectfully disagree with Referee #2. As stated in the title of our manuscript, the objective of our study is to better understand the microphysical processes for stratospheric sulfate aerosol. Although a fraction of the manuscript is about model development, especially the addition of the new stratospheric coarse mode and full chemistry with prognostic OH concentrations, the ultimate aim is to advance our understanding of microphysical processes responsible for the growth of stratospheric sulfate aerosol. The new additions demonstrate the novelty of this work as also mentioned in your comment of #18.

We have heavily revised the sections discussing the physical and chemical processes. In

addition to the description of the magnitude and timing of each process, we now emphasize the roles of interactive OH and the new stratospheric coarse mode in understanding the aerosol processes (see our detailed responses to your comments #12 and #23). We concluded in this study that to better capture observations of stratospheric sulfate burden, aerosol size, AOD and net radiative flux, we need a physical representation of aerosol size distribution while also considering the impact of interactive OH concentrations, which is an important conclusion for the broader scientific community. Therefore, our study fits well within the scope of ACP research articles.

**Response to Comments of Reviewer #1**

**Title:** Size-resolved process understanding of stratospheric sulfate aerosol following the Pinatubo eruption

**General comments**:

The paper is mostly well written with a specific purpose but there are difficulties. The first difficulty is the limited comparisons with observations, just two sets of observations and two figures. Both figures raise questions about the comparison, but little discussion is given to the disagreements. A lot of papers have been published describing the post Pinatubo aerosol from a variety of instruments. Why aren't additional comparisons made with a much wider set of data? The model has pretty fine resolution, so it should not be limited to comparisons to measurements with global coverage. The authors spend a lot of energy comparing the various size distribution modes from the different model configurations, but make no attempt to compare any of these size distributions to observed size distributions. The authors offer no explanation for their limited comparison with observations.

Thanks to the referee for the helpful comments and constructive suggestions. We have revised the manuscript carefully and the point-to-point responses are listed below.

With regards to comparisons with observational data, we have already compared our results against stratospheric sulfate burden from HIRS observations (Figure 4 of revised manuscript) and AOD against AVHRR observations (Figure 10 of revised manuscript). Other sources of observational data are available, such as TOA radiative flux with ERBS and aerosol size comparisons with WOPC and SAGE. However, Brown et al. (2024), another study regarding the simulation of Pinatubo in E3SMv2 (the same model as we used in this work), has mostly already covered these comparisons. Their PA experiment is extremely similar to the MAM4SC experiment in this study, while their SPA experiment is extremely similar to MAM5SC excluding the addition of an independent stratospheric coarse mode. The addition of the new mode or the use of a more complex chemistry scheme do not appear to directly affect model output of TOA radiative flux, which is nearly proportional to stratospheric sulfate burdens. We do not think it is necessary to repeat such comparisons in this work, and instead refer the reader to Brown et al. (2024).

Since a new stratospheric coarse mode is added in this work, we have included a new Figure 6 (see below) in the revised manuscript to compare simulated volume-size distribution against the observations from WOPC following your suggestions. In 1992 and 1993, the coarse mode mass (and volume) for MAM5FC is closer to WOPC observations than MAM5SC.

Furthermore, we have added a global mean AOD anomaly comparison between model simulations and two satellite-derived AOD datasets, AVHRR and GloSSAC (Figure 11 in the revised manuscript, see below). These two observations provided global coverage during the Mt. Pinatubo eruption. Due to limitations in the onboard instruments, AVHRR is more sensitive to the rapid AOD increase caused by volcanic eruptions but becomes less accurate when AOD values fall below 0.01 (Russell et al., 1996; Quaglia et al., 2023). Conversely, GloSSAC measurements become saturated when AOD exceeds 0.15, but are accurate when AOD values are relatively small (Thomason et al., 2018). This justifies using AVHRR and GloSSAC to quantify the upper and lower bounds of the AOD changes caused by Mt. Pinatubo. Consequently, Figure 11 in the revised manuscript has been added to evaluate the performance of different aerosol-chemistry schemes. We assessed the time evolution and e-folding time of the simulated global AOD anomaly against satellite observations (AVHRR and GloSSAC). The MAM5FC and MAM5SC simulations

generally produced reasonable AOD peaks and decay patterns, while MAM4FC and MAM4SC tended to underestimate AOD strength and overestimate the decay rate.

[Figure]

**Figure 6: Comparisons of stratospheric aerosol size distributions from MAM5SC and MAM5FC with observations from WOPC for 1991–1993. WOPC launches are samples taken from the 18 km measurements and matched to the nearest model height and grid cell over Laramie, Wyoming (41.3° N, 105° W).**

[Figure]

**Figure 11: Comparison of AOD anomaly between simulations and observations. The left panel shows the time evolution of monthly AOD anomaly values from simulations and AVHRR and GloSSAC observations, while the right panel shows the time evolution of normalized AOD anomaly values.**

**Major concerns/questions:**
1. 119-120 In terms of size order aren't the modes: Aitken, accumulation, coarse, rather than accumulation first? If so then they should be listed in that order.
   **Response:**
   Thank you for pointing this out. We have revised the text to be ordered by size.

2. Fig. 1 What happens to particles in MAM4 when they exceed Dg_hi (0.48 µm), which is not that large for sulfate particles following Pinatubo? See e.g. Deshler et al., GRL, 1992.
   **Response:**
   In MAM4 renaming is not turned on, therefore the geometric mean diameter is not allowed to exceed Dg_hi. The model will instead increase the aerosol number to maintain conservation of mass while also maintaining a maximum geometric mean diameter. We have added some mention of this in the revised methodology section:
   "In the original MAM4, the accumulation mode was not permitted to rename into coarse mode, and when the accumulation mode's upper limit for aerosol size was reached, continued aerosol growth instead resulted in an increase in aerosol number."

3. 178-179 The parenthetical clause is so long the reader has lost the thread as to what limits the aerosol formation rates.
   **Response:**
   We have restructured this sentence to make it more readable.
   "Most importantly, OH radicals (as well as other oxidants) are prognostically calculated rather than using prescribed values. This allows the model to represent the localized depletion of OH radicals due to the injection of large amount of $SO_2$, bottlenecking aerosol formation rates."

4. Fig. 2 and its discussion. What is the explanation for the aerosol in the Southern Hemisphere, which appears at most longitudes almost simultaneous with the Pinatubo eruption, particularly in MAM5FC? The presence of this aerosol clearly above background should be acknowledged and if possible explained.
   **Response:**
   The issue with aerosol in the Southern Hemisphere is caused by the color bar, because the background values for sulfate aerosol are just slightly larger than the upper limit of the white color. To avoid confusion, we have replotted it with a different color bar (see below).

[Figure]

**Figure 2: Simulated distributions of sulfate aerosol concentrations (kg/kg) at 53 hPa for days 3, 9 and 15 after the eruption of Pinatubo, respectively, for experiments MAM4SC (first row), MAM4FC (second row), MAM5SC (third row), and MAM5FC (bottom row).**

5. 284-286 Why does the smaller geometric standard deviation in MAM5 lead to more persistence? Is it because the particles are smaller in MAM5 compared to MAM4 and therefore less sedimentation? In any case there should be a sentence to describe the physics involved.

**Response:**

The removal rate of aerosols from the stratosphere in E3SM depends heavily on aerosol size, with larger aerosols having a significantly shorter lifespan. This is because the Stokes' settling velocity is roughly proportional to the square of aerosol diameter. With all other factors being identical, when the geometric standard deviation is smaller (i.e. smaller particles are less small, and larger particles are less large), a smaller portion of the aerosol population is large enough to be removed more quickly. However, there is also an important difference in the geometric dry mean diameter. We now explain this in the discussion of Figure 4 and added a reference to the Seinfeld and Pandis textbook (Seinfeld and Pandis, 2016):

"Starting in January 1992, the stratospheric burdens in MAM4 drop off more quickly compared to MAM5-PSA and MAM5FC agrees closely with HIRS until the end of 1992. In MAM4, sulfate in the accumulation mode grows until it reaches its maximum allowed geometric dry mean diameter of 0.48 microns (the Dg_hi value for the accumulation mode). Under such situations, the sulfate aerosol population increases in number (and therefore mass), but the size cannot increase any further. In MAM5-PSA, renaming is permitted, and the transfer from the accumulation mode to the stratospheric coarse mode begins when the geometric dry mean diameter is 0.40 microns (i.e. the Dg_lo value for the stratospheric coarse mode). Upon transfer to the stratospheric coarse mode, certain aerosol growth processes such as coagulation cease (as it is only allowed for the Aitken and accumulation mode in MAM), so the stratospheric coarse mode sulfates in MAM5-PSA tend to grow no larger in size, and the geometric dry mean diameter remains around 0.40 microns. In addition, the reduced geometric standard deviation of MAM5-PSA's stratospheric coarse mode (1.2) relative to MAM4's accumulation mode (1.6) also corresponds to a smaller proportion of super-coarse aerosols that are removed more quickly from the stratosphere, due to the settling velocity being roughly proportional to the second power of the aerosol diameter (Seinfeld and Pandis, 2016). These two reasons are responsible for the sulfate having a longer lifespan in MAM5-PSA than MAM4."

6. Fig. 4 and its discussion. The discussion mostly consists of describing the figure providing specific dates and sulfate burden peaks for the various modes. While perhaps these details are interesting they are all available in the figure for the interested reader. More interesting for the reader would be more discussion of the model / measurement discrepancies. Why do all models except MAM4FC over estimate the peak observed sulfur burden by 20%? Are the HIRS data reliable at the peak or are they suffering a saturation problem? Why does HIRS fall off so much faster than the MAM5 models in 1993? Additional interesting detail would be the range of median sizes involved in the large particle mode.
   **Response:**
   There are several possible reasons for the disagreement. The first is that HIRS results from Baran and Foot (1994) are not particularly accurate outside of the tropical areas due to errors introduced from the background signal (this is acknowledged in their results and discussion section), and as time passes sulfate aerosol is transported towards the poles.
   In general, MAM5FC is able to accurately reflect the observed burden prior to 1993. With respect to the slower fall off in MAM5 models relative to HIRS in 1993, further improvement can be done in our future study. As can be seen in the newly added Figure 6 above, the stratospheric coarse mode burden in MAM5FC consists of particles somewhat smaller than observations, leading to a longer lifespan.

7. Because MAM4 doesn't have a coarse mode and MAM5 has a very narrow accumulation mode, both Figs 4 and 5 show the same thing, no contribution in the accumulation mode (or very little) from MAM5 and no contribution in the coarse mode from MAM4. Why then show the accumulation mode at all? Combine the MAM4 accumulation mode and MAM5 coarse mode into one figure. MAM5 accumulation mode could be included as a dotted line in the coarse mode plot. Then the two models can be more easily compared in terms of sulfate burden carried in the large particle mode.
   **Response:**

The stratospheric coarse mode is unique to MAM5 which does not exist in MAM4, so we would prefer to keep the plots separate in order to emphasize this. However, we have changed the line styles in Figures according to your suggestions here and in your question #16 (we consistently use blue color for MAM4, red for MAM5, solid line for FC, and dashed line for SC). See revised Figure 5 below as an example.

[Figure]

**Figure 5: Simulated stratospheric sulfate burden (in Tg S) divided by mode and latitudinal region. The vertical lines represent the Pinatubo, Cerro Hudson, Spurr and Lascar eruptions respectively.**

8. Fig. 5 Same comments as Fig. 4 but in addition the ordinates should all be the same scale (0-4 Tg S), so the relative contributions from the different latitude zones can be seen directly. Without that the reader immediately wonders about the Southern Hemisphere signal which seems to persist at high levels. The label on the ordinate is wrong. It should be Sulfur burden (Tg S). Label the rows in some other way or describe them in the figure caption. Again combine accumulation and coarse mode into one plot then there will only be three rows. The latitudes of the eruptions should be listed in the figure caption.
**Response:**
Thank you for the suggestion. While using the same scale does have some merits for comparison of magnitude, our original intention is to focus on the aerosol microphysical processes (for example, how long they last in each different mode, etc.). If we set the scale to be identical across the figures, some features would not be very readable, such as Aitken mode with very small burdens. However we have clarified the label of the axis.

9. 331-332 What is the reason for this oscillation, when the sulfur burden is declining everywhere else? A somewhat similar oscillation, but offset, is observed in the Northern Hemisphere. Why does the sulfur burden persist to 1994 in the Southern Hemisphere while it decays everywhere else?

**Response:**

As mentioned in our responses to your question #6, further improvement can be done in our future study. As can be seen in the newly added Figure 6 above, the stratospheric coarse mode burden in MAM5FC consists of particles somewhat smaller than observations, leading to a longer lifespan. If we adjust the size parameters to have larger sulfate particles, we would see larger sedimention along the path of transport and hence smaller burden and oscillation in high latitudes.

10. 341-343 This point would be a lot clearer for the reader if the ordinate scale on all plots was the same. But this statement, "Starting from 1993, the stratospheric burden south of 30 S in MAM5 begins to make up at least half of the total between 80 S and 80 N (about 0.8 Tg S out of the total of 1.7 Tg S)", doesn't make sense. Between 80 S and 80 N includes 30 S – 30 N, so the total sulfur burden in early 1993 for MAM5FC is ~4.1 Tg S.

**Response:**

We prefer to keep the current scales for the reasons described in our responses to your question #8.

Thank you for pointing out the error in numbers. We have corrected it as "(0.8 Tg S out of the total of 4.1 Tg S)".

11. 349-350 Suggest rephrasing to. "The tendencies are the integrals over three-dimensions of: all longitudes, …"

**Response:**

Text has been rephrased as suggested:

"The tendencies are integrated over the three dimensions of: all longitudes, the above-mentioned latitude range, and vertically above the tropopause."

12. Fig. 6 is problematic. There are too many rows making the figure so small that most readers have to blow it up to see it. This could be fixed by breaking it into two figures: the first containing the first 4 rows, the next containing the last 3. Another suggestion is to combine the right and left panels into a single plot with separate ordinates on the right and left. Since the left shows a rate and right shows an accumulation the lines will generally not overlap but rather complement each other. Again the ordinate labels are incorrect. They should be tendency (kg/s) for the left plots and their integrals (or cumulative) (Tg S) for the right. Include the name of the row as a label in each plot. The labels RNMxx are too tied to the inner workings of the model, "renaming". But physically what is happening? The particles are growing to the next largest size distribution mode. If labels were added to the plots to identify them RNMaa could become Aitken->Accumulation mode and RNMasc Accumulation->Coarse mode.

**Response:**

We actually present the units at the top of the figure for both columns, since each column shares the same unit. We have split the figure into three parts. (See below)

Renaming is not the only way for aerosols to transit between modes in E3SM. Please see

[Figure]

**Figure 7: A series of figures for the relevant tendencies for each mode in each experiment in the stratosphere between 80°S and 80°N., in terms of aerosol mass. See Figure 1(a) for description of processes. Positive values represent gained mass from the associated microphysical process from the perspective of the aerosol mode in question. NUCL represents the aerosol mass gain for Aitken mode due to nucleation (i.e., aerosol formation), COAG represents Aitken mode mass loss due to coagulation into the accumulation mode. The left column plots represent the tendency values over time, while the right column represents the cumulative mass change due to the associated microphysical process (i.e. tendency integrated over time). The vertical lines represent the Pinatubo, Cerro Hudson, Spurr and Lascar eruptions respectively.**

[Figure]

**Figure 8: Tendency plots for condensation in the different aerosol modes, similar to those for nucleation and coagulation. Positive values represent mass transition from the gaseous phase to the aerosol.**

[Figure]

**Figure 9: Tendency plots for renaming, i.e. the transition of mass between aerosol modes due to the aerosol population reaching a prerequisite size, as described in Figure 1(a). RNMaa represents the mass transition from the Aitken mode to the accumulation mode (renaming is only possible in this direction, and cannot occur in the reverse), and the negative value represents the mass loss of the Aitken mode. RNMasc represents the mass transition from the accumulation mode to the stratospheric coarse mode (also only possible in this direction), and the positive value represents the mass gain of the stratospheric coarse mode.**

13. Fig 6 e, f) shows the growth from Aitken to accumulation mode presumably by several processes including growth by condensation and coagulation, correct? If that is the case then why are the ordinate scales on COAG so much larger than for RNMaa? Both are showing mass loss rate and total mass lost. It seems it should be the other way around with COAG less than RNMaa. Why is this one process coagulation singled out for a special plot?
**Response:**
In the model, condensation, coagulation and renaming are each different aerosol processes. Condensation/evaporation represents the gas-aerosol mass exchange, e.g. transition from $H_2SO_4$ gas to sulfate aerosol. Coagulation is the process of multiple smaller particles colliding into each other to form larger particles. Renaming is not a "real" physical process and is instead an internal calculation within the model where particles that have grown sufficiently large are moved from one aerosol mode to another. Coagulation and renaming are two separate ways for aerosols to transit from the Aitken mode to accumulation mode; coagulation is not a subset of renaming or vice versa.

Because COAG and RNM are separate mechanisms, we think it is better to keep them separate in figures and discussions. We have also added a more detailed explanation in the methodology section of what these processes represent:

"Renaming is not a physical process per se, but transfers aerosols from one mode to another as a result of microphysical processes such as condensation and coagulation. The rate of renaming is controlled by the microphysical processes. It acts to increase aerosol size (not in both directions). We have considered seven different microphysical processes regarding the

sulfate aerosol in the manuscript. The following explanation has been added to the methodology section to help the reader understand the differences between them:

- NUCL, or nucleation, is responsible for the formation of aerosols starting from the Aitken mode (nucleation mode aerosols are not represented in MAM).
- COAG, or coagulation, represents two smaller particles colliding to form a larger particle. MAM considers the intramodal and intermodal coagulation of the Aitken, accumulation and primary carbon modes. Intramodal coagulation reduces the number of the mode but leaves the mass unchanged. Intermodal coagulation between these modes and coarse mode is not considered in MAM.
- AitkenCond, or condensation in the Aitken mode, leading to aerosol mass increase in the Aitken mode population. No matter how much mass is gained from condensation, the aerosols cannot grow into the accumulation mode directly, it MUST undergo renaming.
- AccumCond and StratoCoarseCond, condensation for the accumulation mode and stratospheric coarse mode, work similarly.
- RNMaa, or renaming from the Aitken to accumulation mode. Periodically, the size of the Aitken mode aerosol population is checked. If a specified threshold is reached (due to mass increase from the condensation and coagulation processes), then aerosol number and mass are transferred from the Aitken mode to the accumulation mode.
- RNMasc, renaming from the accumulation mode to stratospheric coarse mode. No other microphysical process is allowed to produce stratospheric coarse mode sulfates, so this is their only source."

14. Fig 6 i – n) Condensation? Why are these processes now called condensation? Condensing from what? Weren't these earlier called renaming, which is also not that helpful or descriptive. Isn't this particle growth from one mode to another? Ordinate problems again. What is on the left and right ordinates? Is it again rate (kg/s) and cumulative mass (Tg S)? The reader doesn't know and the figure caption does not help.
**Response:**
See our responses to your question #13. We have now clarified the terminology in figure captions.

15. 351-415 Again the figure discussion consists primarily of describing the figure, pointing out maximum values and dates when they occur. Relatively little is describing what can be learned from the figure which is the importance of the figure. Here and elsewhere, if these dates and amounts are particularly important organize them into a table. Then they could really be compared. It is not clear how listing them in the text helps the reader.
**Response:**
Following the reviewer's suggestion, we have heavily revised the discussion of this figure to better explain the relationships between these processes:

"With respect to nucleation, Figure 7(a) shows the magnitude of the tendency for the formation of Aitken mode aerosols through the nucleation of gaseous $H_2SO_4$ (NUCL) is generally higher in the FC experiments compared to the SC experiments, which leads to the higher Aitken mode sulfate concentrations in FC than in SC as shown in Fig. 4(b). Although one would expect SC to have faster nucleation without the bottleneck of OH concentrations,

new particle formation by nucleation competes against aerosol growth by condensation. Condensation is stronger in the SC experiments than FC for the first five months after the eruption, predominantly in the accumulation mode condensation for MAM4 (Figs. 8 (c)(d)) and in the stratospheric coarse mode for MAM5-PSA (Figs. 8 (e)(f)). This leads to competition with the nucleation process, so FC has stronger nucleation than SC. Due to the higher OH concentrations in the SC experiments, the nucleation process begins and ends earlier in the SC experiments compared to FC. Considering the cumulative mass changes depicted in Fig. 7(b), the curves for the SC experiments flatten out (i.e., the microphysical process in SC is mostly concluded) earlier than those for FC.

With respect to coagulation, Figs. 7(c)(d) show the aerosol mass loss rates (negative values) of the Aitken mode due to coagulation into the accumulation mode (COAG). The magnitude of the coagulation generally follows that of the nucleation process, as the strength of the coagulation process is proportional to Aitken mode aerosol number and mass.

With respect to the condensation processes, Figs. 8(a)-(f) show the condensation processes for each of the three modes. Fig. S6 in the supplementary shows the sum of condensation processes across the three modes for reference. Across the first five months after the eruption, for Aitken mode condensation, FC has stronger condensation tendencies than SC corresponding to the larger burden of Aitken mode aerosols seen in Figure 4(b). For the accumulation mode, condensation is stronger for MAM4 than MAM5-PSA likewise corresponding to the very small accumulation mode burden in MAM5-PSA (Fig. 4(c)). For the stratospheric coarse mode, MAM5SC has a much stronger condensation tendency than MAM5FC corresponding to the former having a higher stratospheric coarse mode burden (Fig. 4(d)). It should be noted that MAM4SC similarly has stronger condensation tendency for the accumulation mode than MAM4FC, as the sulfate does not grow beyond the accumulation mode. After five months, condensation concludes in all of the experiments except for MAM4FC, due to delays from its OH bottleneck.

Figs. 9(a)(b) show the aerosol mass loss rate of the Aitken mode due to renaming into the accumulation mode (RNMaa). Similar to coagulation above, the strength of RNMaa is also dependent on the preceding processes of nucleation and condensation, which provide the large Aitken mode particles necessary for renaming. Since nucleation and Aitken mode condensation is stronger in FC than the SC experiments, this is also true for RNMaa.

Figs. 9(c)(d) represent the aerosol mass gain rate of the stratospheric coarse mode due to renaming from the accumulation mode (RNMasc). Similar to RNMaa, the relative strength is determined by the prior steps, here being RNMaa, COAG and AccumCond, leading to MAM5FC being stronger than MAM5SC.

Overall, SC has stronger condensation processes (sum for all modes) and weaker nucleation (i.e. initial formation of Aitken mode sulfates) relative to FC over the first five months, again showing the competition between condensation and nucleation for $H_2SO_4$. Because coagulation and renaming are dependent on nucleation and condensation, FC is stronger than SC for coagulation and renaming. With respect to the differences in processes between MAM4

and MAM5-PSA, MAM4 has stronger nucleation, coagulation and renaming processes due to weaker competition from condensation."

16. Figs. 4-8, 11 The plotting for these figures could be made much more intuitive, so the reader doesn't constantly have to refer to the legend to remember which is which. It would be quite easy to do. Use one color for MAM4 and one for MAM5, then one line style (e.g. solid) for FC and another (dashed or dotted) for SC. Then each figure can be immediately understood without referring to the legend but once.
    **Response:**
    Thank you for the suggestion. We have changed the line styles in these figures to consistently use blue color for MAM4, red for MAM5, solid line for FC, and dashed line for SC).

17. Figs 7 and 8 suffer from the same problem as Fig. 6. There are too many panels and they are too small. What are Figs. 7 and 8 adding to what we learned from Fig. 6? Are all these rows necessary? Which ones are the most informative?
    **Response:**
    Figures 7-9 (previously Fig. 6) indeed has too many panels to show processes for all the modes. It has now been split into three figures, though all of the rows have been retained as they are necessary for the discussion.

18. 420-430 In fact the discussion of Figs. 7 and 8 acknowledges that not much new is added. "The same patterns as above apply between 30 S and 30 N. Above 30 N and below 30 S the same signal from the Pinatubo is still present, though slightly delayed due to the time that it took for the aerosol to transport poleward. Signals from other eruptions (Spurr and Lascar) are also present" Then a few interesting differences are discussed. Just show the interesting panels and combine Figs 7 and 8 into one figure with the few interesting panels.
    **Response:**
    We had discussions regarding condensation across different latitudinal regions in the text. However, following your suggestions, we have moved the original Figs. 7 and 8 to the supplementary.

19. 503 condensation? Same questions as above. What does this mean?
    **Response:**
    See our responses to your question #13.

20. 503-505 What is the reason for quoting these numbers? How will the reader use such information rather than the already stated comparison about the difference rates? Too detailed.
    **Response:**
    We have heavily revised the conclusion section following your comments and that of the handling editor's.

    "In this work, we have implemented a fifth stratospheric coarse mode (MAM5-PSA) on top of the original four-mode version Modal Aerosol Module (MAM4) used in E3SMv2. We also consider a more complex "full chemistry (FC)" set-up, which includes the addition of a series of chemical reactions which allow the simulation of OH radical replenishment compared to "simple chemistry (SC)" with prescribed OH concentrations. We have carried out a series of

simulations of the Pinatubo eruption for the time period of 1991-1993 to better understand the aerosol microphysical processes including nucleation, coagulation and condensation. We have also compared model simulation results with measurements of stratospheric sulfate burden, aerosol size, AOD and net radiative flux.

[revised manuscript text omitted]

21. 506 Here COAG is separated from RNMaa, but aren't both processes doing the same thing? There is only the transition from Aitken to accumulation. Is it important how it happens? If so why isn't that mentioned earlier?
   **Response:**
   See our responses to your question #13.

22. 509 deposition? Does this mean sedimentation out of the stratosphere? Generally deposition refers to losing aerosol due to contact with a surface.
   **Response:**
   "deposition" has been changed to "sedimentation".

**Response to Comments of Reviewer #2**

**Title:** Size-resolved process understanding of stratospheric sulfate aerosol following the Pinatubo eruption

**General comments**:

I think the article is well written and a nice advancement to stratospheric aerosol modeling but may be better suited in its current form for a journal such as Geoscientific Model Development. While the differences in aerosol loading due to model configuration are made clear, it is left to the reader to interpret how this improves understanding of atmospheric chemistry or physics. The authors provide in-depth discussion on the relative importance of coagulation, nucleation and renaming/growth in the model, but have little discussion on the physical processes this may help resolve. Similarly, few comparisons are made with measurements for AOD, particle size or radiative flux, with no discussion given to possible sources of disagreement, or what implications these results may have for observations (e.g. the assumptions going into the HIRS results used here). Personally, I think addressing any of these points would help expand the applicability of the paper to a more general audience.

Thanks to the referee for the helpful comments and constructive suggestions. We have revised the manuscript carefully and the point-to-point responses are listed below.

As shown by the title of the manuscript, this work aims to provide in-depth discussion on the relative importance of coagulation, nucleation and renaming/growth in stratospheric sulfate formation.

With regards to comparisons with observational data, we have already compared our results against stratospheric sulfate burden from HIRS observations (Figure 4 of revised manuscript) and AOD against AVHRR observations (Fig. 10 of revised manuscript). Other sources of observational data are available, such as TOA radiative flux with ERBS and aerosol size comparisons with WOPC and SAGE. However, Brown et al. (2024), another study regarding the simulation of Pinatubo in E3SMv2 (the same model as we used in this work), has mostly already covered these comparisons. Their PA experiment is extremely similar to the MAM4SC experiment in this study, while their SPA experiment is extremely similar to MAM5SC excluding the addition of an independent stratospheric coarse mode. We therefore refer the reader to Brown et al. (2024) for TOA flux comparisons. For reference, it can be seen from the net TOA flux comparison in Brown et al. (2024) that from July 1991 to the end of 1992, observed net radiative flux anomalies are within the range of -3.8 to -1 W/m$^2$. Their E3SMv2-PA's simulated values are in the range of -2 to +1.5 W/m$^2$, while for our MAM5FC experiment, they are in the range of -2.8 to -1.3 W/m$^2$. In terms of net radiative flux anomaly, MAM5FC agrees more closely with the ERBS observations.

[Figure]

**Figure R1: screenshot of Figure 4 in Brown et al. (2024), a comparison of their E3SMv2-PA simulated net radiative flux anomalies (solid blue line) and ERBS observations (solid black line).**

[Figure]

**Figure 13 from the manuscript, representing net radiative flux anomalies of the four experiments.**

Since a new stratospheric coarse mode is added in this work, we have included a new Figure 6 to compare simulated volume-size distribution against the observations from WOPC following your suggestions. It can be seen that MAM5FC did better capture the coarse mode volume (or mass) of sulfate aerosol in 1992 and 1993.

We added a global mean AOD anomaly comparison between model simulations and satellite-derived AOD datasets, AVHRR and GloSSAC, which provided global coverage during the Mt. Pinatubo eruption. AVHRR is more sensitive to rapid AOD increases caused by eruptions but becomes less accurate for AOD values below 0.01, while GloSSAC is accurate at lower AOD

values but becomes saturated above 0.15. These observations help quantify the bounds of AOD changes from Mt. Pinatubo. Figure 9 in the revised document was added to evaluate different aerosol-chemistry schemes. The MAM5FC and MAM5SC simulations showed reasonable AOD peaks and decay patterns, while MAM4FC and MAM4SC tended to underestimate AOD strength and overestimate the decay rate.

**Major concerns/questions:**

1. Line 50-65: This geoengineering section seems a bit out of place to me. I'm sure this work has implications for geoengineering studies, but no indication of exactly what those may be is provided. I recommend clarifying the link to this work or removing this paragraph. Perhaps the geoengineering discussion and the link to this work would be better placed in the discussion/conclusion?
   **Response:**
   Geo-engineering is commonly cited as a motivation into the simulation of volcanic aerosols/stratospheric sulfates, e.g. Mills et al. (2017) introduction and Tilmes et al. (2022).

2. Line 415: "MAM4 also generally has stronger nucleation and coagulation processes than MAM5." From Figure 6 it isn't clear whether MAM4FC has greater coagulation tendencies for physical reasons or if it is just due to the increased NUCL. This is discussed in the conclusion of the paper (Line 503), but I think should be mentioned here. I would suggest rewording to something like: "MAM4 also generally has stronger nucleation than MAM5, and due to these higher concentrations, increased coagulation processes as well" Or, if there are other reasons for increased COAG then this should also be discussed.
   **Response:**
   Thank you for the suggestion; this section has been heavily rewritten in its entirety in response to feedback from the handling editor, and this portion no longer exists.

3. Line 490: "large differences in both the temporal variations and the spatial distributions of sulfate concentrations" It is difficult to tell from Figures 7/8, but in Figures 9/10 there doesn't appear to be much change in spatial distribution. Both MAM4 and MAM5 show large increase in the tropics and later transport to the NH. Some expansion on the spatial differences the authors are referring to would be welcome.
   **Response:**
   This is primarily referring to Fig. 3, where SC experiments had sulfate concentrations at a noticeably higher altitude than FC. We've changed the word "spatial" to "vertical".

4. Line 521-522: If the use of full chemistry and MAM5 helped improve agreement with AVHRR, why is the TOA flux more comparable to Brown (2024) and Mills (2017) results than the MAM4 version? Is this related to the geometric standard deviation that was used?
   **Response:**
   The fundamental changes made to the aerosol module are similar across MAM5 in this work, the altered MAM4 scheme in Brown et al. (2024), and the altered scheme in Mills et al. (2017): they all allow sulfate to rename into coarse mode and adjusting some of the coarse mode parameters (e.g., reduced the geometric standard deviation of the coarse mode in their altered MAM4 to that as used in the stratospheric coarse mode in MAM5 of this work) to better fit observational data. On the contrary, MAM4 in this work does not allow sulfate to rename into

coarse mode (i.e., continued increase in sulfate mass leads to an increase in the accumulation mode number concentration, rather than a transition from accumulation mode mass to coarse mode mass). As a result, the TOA flux from MAM5 agrees closer to Brown (2024) and Mills (2017) than that from MAM4, and this is the expected result.